# Reversible modulation of superconductivity in thin-film NbSe$_2$ via plasmon coupling

Guanghui Cheng ®[1,2] ✉, Meng-Hsien Lin[3], Hung-Ying Chen[3], Dongli Wang[4], Zheyan Wang[1], Wei Qin ®[1] ✉, Zhenyu Zhang ®[4,5] & Changgan Zeng ®[1,4,5] ✉

In recent years, lightwave has stood out as an ultrafast, non-contact control knob for developing compact superconducting circuitry. However, the modulation efficiency is limited by the low photoresponse of superconductors. Plasmons, with the advantages of strong light-matter interaction, present a promising route to overcome the limitations. Here we achieve effective modulation of superconductivity in thin-film NbSe$_2$ via near-field coupling to plasmons in gold nanoparticles. Upon resonant plasmon excitation, the superconductivity of NbSe$_2$ is substantially suppressed. The modulation factor exceeds 40% at a photon flux of $9.36 \times 10^{13} \, \text{s}^{-1} \text{mm}^{-2}$, and the effect is significantly diminished for thicker NbSe$_2$ samples. Our observations can be theoretically interpreted by invoking the non-equilibrium electron distribution in NbSe$_2$ driven by the plasmon-associated evanescent field. Finally, a reversible plasmon-driven superconducting switch is realized in this system. These findings highlight plasmonic tailoring of quantum states as an innovative strategy for superconducting electronics.

The strong coupling between superconductors and electromagnetic waves not only reveals emergent material phases hidden beneath superconductivity[1,2], but also leads to control approaches essential for the development of dissipationless superconducting circuits, including cryogenic switches[3], superconducting transistors[4] and tunable qubits[5]. Lightwave, in particular, can be a noninvasive and remote knob to manipulate superconducting devices. Its ultrafast and reversible characteristics are crucial for applications that require dynamic modulation or switching between different electronic states. So far, extensive efforts have yielded promising results in using light to control superconductivity[1–4,6–9]. For example, intense infrared or THz pulses have been employed to drive materials into nonequilibrium superconducting states[2,6,7]. Photo-induced superconductivity phase transition has been realized through photochemical processes[4]. However, due to the weak light-matter interactions of superconducting materials, finding a universal

approach for efficient and reliable control over superconductivity remains challenging.

Moreover, it is of fundamental significance to investigate the interaction mechanism between light and superconducting materials. While various mechanisms have been proposed, such as the modulation of electron-phonon interactions[6,10], the injection of quasiparticles[9,11], and the manipulation of the superconducting order parameters[2], further experimental and theoretical studies are highly desired to develop a comprehensive understanding of the underlying mechanisms.

A promising solution for bridging light with superconductors is via near-field plasmon coupling at the heterointerface. The extreme confinement of light into the nanoscale allows for strong interactions of plasmon modes with various quasiparticles, such as excitons[12,13] and phonons[14,15], which facilitate enhanced light-matter interactions[16], energy transfer/relaxation processes[17], noncollinear

[1]CAS Key Laboratory of Strongly Coupled Quantum Matter Physics, and Department of Physics, University of Science and Technology of China, Hefei, China. [2]Advanced Institute for Materials Research (WPI-AIMR), Tohoku University, Sendai, Japan. [3]MetaSERS TECHNOLOGY Corp., Hsinchu, Taiwan. [4]International Center for Quantum Design of Functional Materials (ICQD), Hefei National Research Center for Physical Sciences at the Microscale, University of Science and Technology of China, Hefei, China. [5]Hefei National Laboratory, Hefei, China. ✉e-mail: cheng.guanghui.c2@tohoku.ac.jp; qinwei5@ustc.edu.cn; cgzeng@ustc.edu.cn

optical phenomena[18]. More strikingly, plasmons have exhibited quantum behaviors, showing their ability to mediate photon entanglement during photon-plasmon-photon conversion[19,20] as well as to enhance the quantum coherence of electrons through delicate electron-plasmon coupling[21]. Nevertheless, the specific manner in which surface plasmons modify coherent Cooper pairs has yet to be demonstrated.

In this study, we combine a typical two-dimensional (2D) superconductor NbSe₂ with plasmonic gold nanoparticles (AuNPs) in a hybrid device, as illustrated in Fig. 1a. We employ hBN (thickness ~5 nm) as a thin insulating layer separating AuNPs and NbSe₂ to avoid charge transfer between them and to protect NbSe₂ from degradation. To avoid degradation of NbSe₂, pre-patterned gold electrodes are initially fabricated on silicon wafers, and the subsequent exfoliation and pick-stack processes of NbSe₂ and hBN flakes are both performed in an argon glovebox. Figure 1b shows the optical image of device A1 with a trilayer NbSe₂. The scanning electron microscope (SEM) image shows the monolayer of well-organized AuNPs on top of hBN/NbSe₂. Further information regarding the device fabrication and transfer of AuNPs is provided in Methods and Supplementary Note 1, respectively. Figure 1c shows the optical absorption spectra of the AuNPs (green curve), revealing a pronounced plasmon resonance at 567 nm.

Light with two typical wavelengths (532 nm and 1064 nm) used in this study is denoted by the black arrows. In the following discussion, we refer to the illumination of 532(1064)-nm light as the on(off) resonant plasmon excitation.

## Results and Discussion

### Modulation of $T_c$ in the hybrid NbSe₂-AuNPs devices

Figure 1d, f show typical results of the superconducting transitions in a trilayer NbSe₂ device A1 under on(off)-resonant plasmon excitation. Notably, the modulation of superconductivity by on-resonant plasmon excitation is much stronger than that under off-resonant plasmon excitation. We note that the maximum photon fluxes of on/off resonance are different due to the maximum output of the lasers we used. Typical R-T curves at the same photon flux ($4.68 \times 10^{13}$ s⁻¹mm⁻²) are shown in Fig. 1e, g for comparison. With the dark case as the reference, the change of the transition temperature $T_c$ of NbSe₂ is negligible under off-resonant plasmon excitation. In contrast, $T_c$ decreases from 4.1 K (dark case) to 3.7 K under on-resonant plasmon excitation. Note that at the maximum photon flux of $9.36 \times 10^{13}$ s⁻¹mm⁻² in Fig. 1d, $T_c$ drops to 2.9 K with a modulation factor $\frac{|T_c - T_{c0}|}{T_{c0}}$ up to 41% ($T_{c0}$ is the critical temperature for the dark case). An additional resistance drop

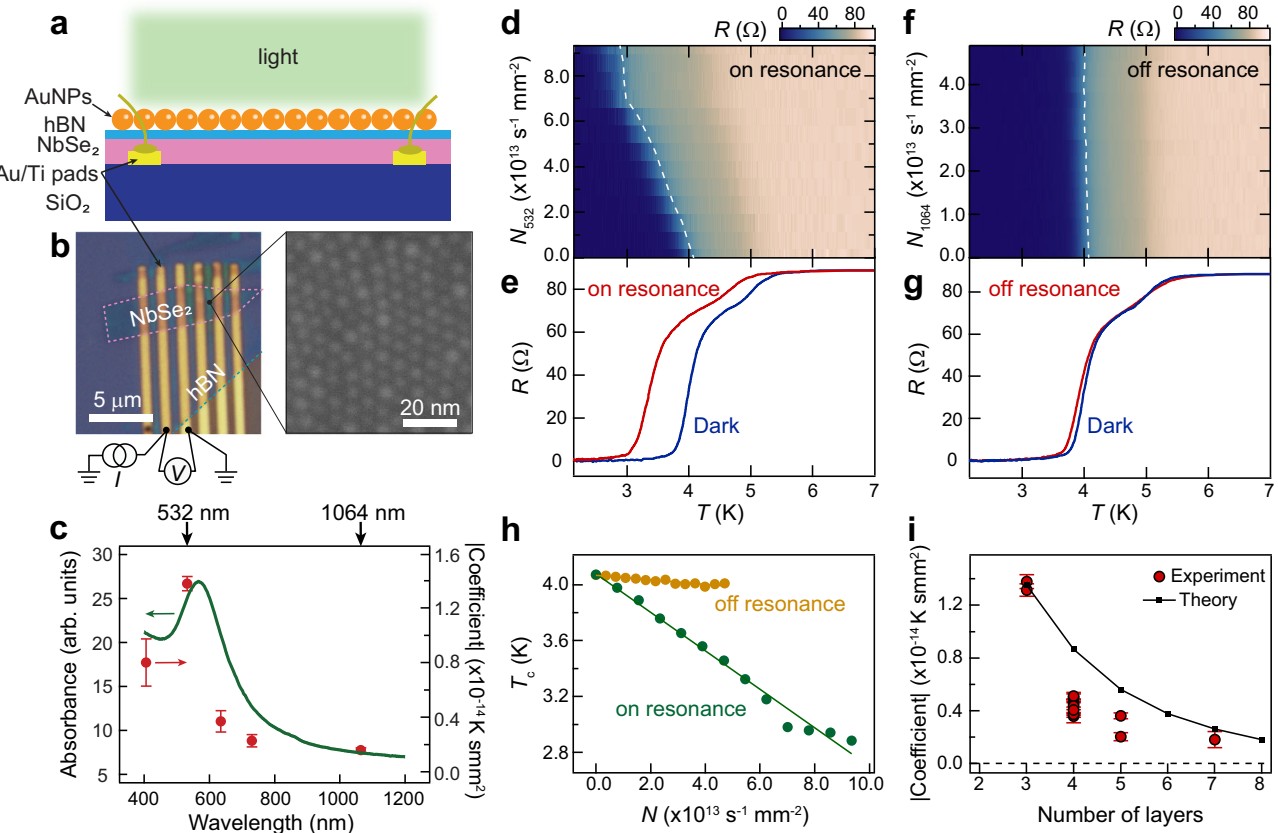

**Fig. 1 | Hybrid device of plasmonic superconductor and the superconductivity modulation by plasmon excitation. a** Schematic of the gold nanoparticles (AuNPs)/hexagonal boron nitride (hBN)/NbSe₂ device under light illumination. **b** Optical image of a typical device A1 with a two-terminal measurement configuration and scanning electron microscope (SEM) image of the AuNPs organized on top of hBN/NbSe₂. The pink and cyan dashed lines outline the NbSe₂ and hBN flakes. **c** Left axis: Optical absorption spectra of the AuNPs on a quartz substrate. The black arrows denote the on (off)-resonant plasmon excitation by 532(1064)-nm light. Right axis: $T_c$ modulation coefficient versus illumination wavelength. The error bars are obtained by the linear fittings in Supplementary Fig. 4. **d–f** Sample resistance $R$ of device A1 as functions of temperature $T$ and photon flux $N_{532}$ ($N_{1064}$) under on(off)-resonant plasmon excitation by 532(1064)-nm light illumination. A contact

resistance $R_c = 54.5$ Ω has been deducted. The white dashed lines denote the superconducting transitions where $R$ reaches 50% of the normal state resistance. **e–g** Typical R-T curves for dark and on/off resonance cases (at the same photon flux of $4.68 \times 10^{13}$ s⁻¹mm⁻²). **h** Superconducting critical temperature $T_c$ versus photon flux $N$ under on(off)-resonant plasmon excitation. $T_c$ is defined by the temperature at which $R$ reaches 50% of the normal state resistance. Linear fittings yield $T_c$ modulation coefficients of $-1.38 \times 10^{-14}$ Ksmm² and $-0.16 \times 10^{-14}$ Ksmm² for on- and off-resonant plasmon excitations, respectively. **i** $T_c$ modulation coefficient as a function of the number of layers of NbSe₂. The red circles and black squares are experimental data and theoretical predictions, respectively. The experimental data (red circles) are collected from twelve NbSe₂ devices with different thicknesses. The error bars are obtained by the same method as in **c**.

can be observed ~5 K possibly due to the nonuniformity of the $NbSe_2$ sample, caused by the nanofabrication processes and the transfer of AuNPs.

In principle, without AuNPs, intense light pulses with photon energies larger than the superconductivity gap can induce quenching of superconductivity through the quasiparticle excitations[8,22]. However, further control experiments on a trilayer $NbSe_2$ device without depositing AuNPs do not show noticeable modulation of superconductivity (see the comparison under similar photon fluxes in Supplementary Fig. 3), possibly due to the weak continuous-wave light intensity and the low absorption of thin-film $NbSe_2$ in this work.

The extracted values of $T_c$ are summarized in Fig. 1h as a function of the photon flux $N$. The dropping of $T_c$ roughly follows a linear dependence on the photon flux $N$. Linear fittings yield $T_c$ modulation coefficients of $-1.4 \times 10^{-14}$ Ksmm$^2$ and $-0.3 \times 10^{-14}$ Ksmm$^2$ for on- and off-resonant plasmon excitations, respectively. The modulation coefficients at five selected illumination wavelengths (406, 532, 635, 730, and 1064 nm) are summarized in Fig. 1c (detailed data are shown in Supplementary Fig. 4). Remarkably, the superconductivity modulation coefficient roughly follows the absorbance of AuNPs, reaching maximum close to the resonance peak of plasmon excitation. These observations indicate that the resonant plasmon excitation of AuNPs plays an essential role in modulating the superconductivity of $NbSe_2$. We note that quantifying the light intensity in terms of power does not alter our observations (see the replotted $R$-$T$ curves and modulation coefficient in terms of power in Supplementary Figs. 5, 6).

Moreover, the observed plasmon-induced superconductivity modulation depends highly on the thickness of $NbSe_2$, as shown in Fig. 1i. Each data point corresponds to an $NbSe_2$ device (thicknesses are confirmed by the atomic force microscopy and optical contrast). The thicknesses of hBN separating AuNPs and $NbSe_2$ have been deliberately chosen to be approximately 5 nm. As the thickness of $NbSe_2$ increases, the modulation coefficient drops towards zero, indicating the quick quenching of the plasmon-induced modulation effect when getting far away from the AuNPs. The nanometer-scale thickness dependence suggests that the quenching of superconductivity cannot be simply attributed to the sample heating effect (typical thermal penetration depth is at micron scale[23]). Instead, such thickness dependence points to a near-field-related mechanism localized at the nanoscale surface/interface[24] of the AuNPs/hBN/$NbSe_2$ heterostructure. We note that the observed thickness dependence is not caused by the skin effect of superconductors because the penetration depth of $NbSe_2$ is ~250 nm[25], which is much larger than the thickness of thin-film $NbSe_2$ (a few layers).

A phenomenological model is developed to understand the observed modulation of superconductivity by plasmon (See details in Supplementary Note 2). The plasmon excitations in AuNPs transform the propagation light into an evanescent field[26] that decays exponentially away from the AuNPs into the $NbSe_2$ layers (The field cannot be fully screened by the 2D superconductor due to the relatively large penetration depth[25]). Note that the strength of plasmon-induced evanescent field close to $NbSe_2$ layers is at the same level as the far field of propagation light (see the Finite Difference Time Domain (FDTD) simulation in Supplementary Fig. 7), which cannot explain the disparity between the substantial and negligible modulations of superconductivity with and without AuNPs. We emphasize that the key factor for the plasmon-enhanced light-matter interaction in this work is not the enhancement of the field strength, but the higher efficiency of the evanescent field-electron coupling than the propagation light-electron coupling. Essentially, the evanescent field can induce electron-hole excitations[27] in a more efficient way due to its larger and broader momentum distribution[28,29]. These plasmon-excited hot carriers rapidly relax energy via electron-electron scatterings and electron-phonon scatterings[22,30–32] within tens to hundreds of picoseconds[9,31,33], quickly driving the system into a dynamical

equilibrium state upon continuous light illumination. The plasmon-induced electron-hole excitations and the associated phonon population with energies surpassing the superconducting gap contribute to Cooper pair breaking in the superconducting state, leading to the suppression of superconductivity in $NbSe_2$. To give a quantitative analysis, we assume that this dynamical equilibrium state can be modeled by the redistribution of electrons characterized by an effective layer-dependent quasiparticle temperature $T_l^* = T_1^* \exp[(1-l)\eta]$, where $T_l^*$ is the effective temperature of the $l$-th layer and $\eta$ denotes the decay coefficient of the plasmon-induced evanescent field. Based on these assumptions, we calculate the superconducting critical temperatures via solving the linearized gap equation (see detailed calculations in Supplementary Note 2).

Our theoretical results show that the initial stage of $T_c$ suppression exhibits a nearly linear behavior (Supplementary Figs. 8a, 9a), consistent with experimental observations in the trilayer $NbSe_2$ (Fig. 1h). Moreover, by choosing the model parameters from the fittings of trilayer $NbSe_2$ (Supplementary Fig. 9a), our theoretical analysis predicts that the $T_c$ modulation coefficient decays exponentially upon increasing the thickness of $NbSe_2$ (black rectangles in Fig. 1i), consistent with the experimental data (red circles in Fig. 1i and Supplementary Fig. 9b). Therefore, we conclude that the coupling between $NbSe_2$ and the significant evanescent field induced by the resonant excitation of plasmons drives the electron distribution in $NbSe_2$ out of thermal equilibrium, resulting in the suppression of superconductivity.

## Characteristics of plasmon-coupled superconductivity

Next, we investigate the basic characteristics of superconductivity in the hybrid AuNPs/hBN/$NbSe_2$ device under resonant plasmon excitation. Figure 2a shows the voltage-current ($V$-$I$) characteristics on a logarithmic scale at various photon fluxes, taken at a temperature of 2.2 K (the full range data on a linear scale are provided in Supplementary Fig. 10). As photon flux increases, the $V$-$I$ characteristics gradually transition from zero-resistance supercurrent to a linear dissipation state. At the critical photon flux $N_{BKT} = 6.24 \times 10^{13}$ s$^{-1}$mm$^{-2}$, the $V$-$I$ characteristics obey the universal scaling relation $V \sim I^3$ (marked by the dashed line). This behavior suggests the presence of Berezinskii-Kosterlitz-Thouless (BKT) transition, which was previously reported in the thermal-driven evolution of 2D superconductivity at finite temperatures[34,35]. Although the BKT transition typically occurs in materials with large sheet resistance[36], this is not an exclusive requirement considering the relatively low sheet resistance in our case ($R_N^\square \sim 468\Omega$) and in other 2D superconductors[35,37,38]. The BKT transition is further validated by the temperature induced vortices/antivortices unbinding model[38], providing reasonable fittings of the observed $R$-$T$ curves using the Halperin-Nelson formula[34] (Supplementary Fig. 11). In the present system, the superconducting phase consists of bound vortex-antivortex pairs at low temperature[36]. The presence of proximal plasmon resonance reduces the critical temperature $T_c$, driving the system to approach the BKT phase transition of unbinding vortex-antivortex, resulting in the onset of dissipation.

The differential conductance spectra, represented by d$I$/d$V$ versus $V$, are derived through numerical differentiation of the results in Fig. 2a (Supplementary Note 3). The two symmetric d$I$/d$V$ peaks arise from the normal electrons transmitting into the electron and hole branches of the quasiparticle spectrum in the superconductor[39]. Therefore, the separation between the two peaks corresponds to the superconducting gap $2\Delta$. Utilizing the Blonder-Tinkham-Klapwijk model[39] for quantitative analysis (Supplementary Note 3), the superconducting gap size $\Delta$ as a function of photon flux is depicted in Fig. 2b. The critical current $I_c$ is extracted from the superconducting-to-normal transitions in Fig. 2a and summarized as a function of photon flux in Fig. 2c. Both $\Delta$ and $I_c$ exhibit linear decreases with increasing the photon flux, being consistent with the results of $T_c$ shown in Fig. 1h. To determine the role

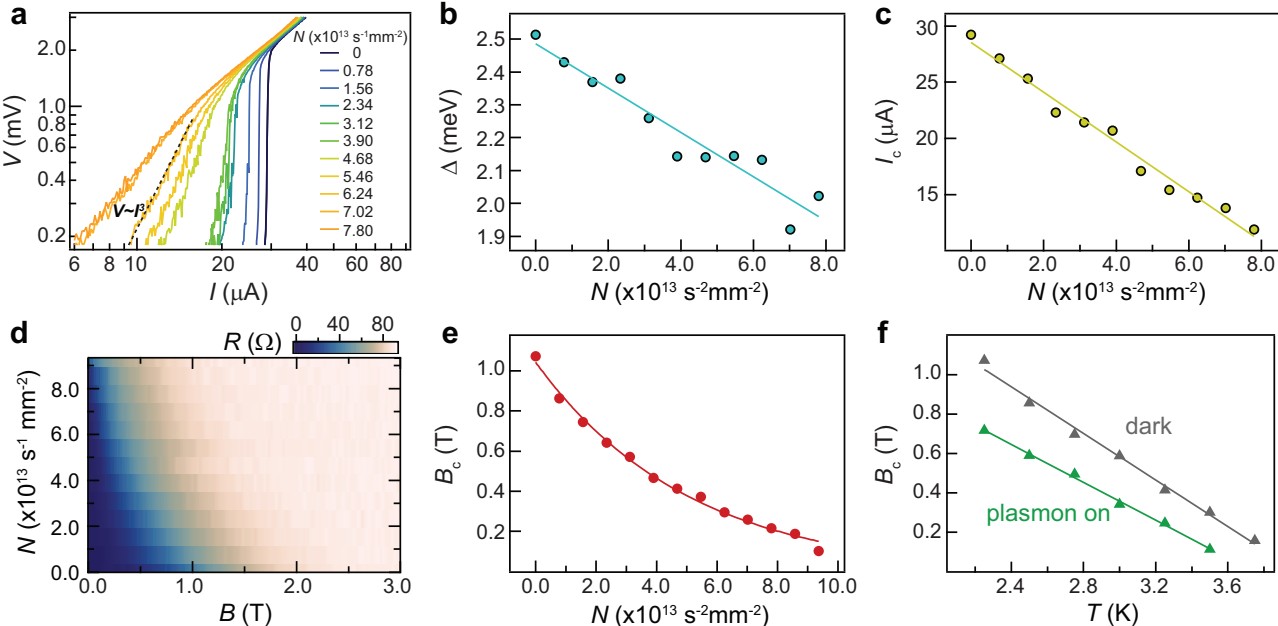

**Fig. 2 | Characteristics of superconductivity under resonant plasmon excitation. a** Voltage–current (*V-I*) characteristics of device A1 (AuNPs/hBN/NbSe₂) on a logarithmic scale at various photon fluxes. The black dashed line denotes the expected $V$ - $I^3$ behavior for the BKT transition. **b, c** Photon flux dependence of the superconducting gap size Δ and critical current $I_c$. Δ is extracted by fitting the differential conductance spectra, d*I*/d*V* versus *V*, within the Blonder-Tinkham-Klapwijk model (Supplementary Fig. 12). $I_c$ is extracted from the superconducting-to-normal transitions in (**a**). The linear fittings are guides to the eye. **d** Sample resistance as functions of perpendicular magnetic field *B* and photon flux *N*. **e** Photon flux dependence of the critical field $B_c$, extracted from (**d**). The power fitting is the guide to the eye. All the data in **a-e** are taken at a temperature of 2.2 K. **f** Temperature dependence of the critical field $B_c$ without and with plasmon excitation ($N = 1.56 \times 10^{13}$ s⁻¹mm⁻²), extracted from the mappings of resistance versus temperature and magnetic field in Supplementary Fig. 15. The linear fittings give Ginzburg–Landau coherence lengths $\xi_{GL}$ of 11.8 and 13.5 nm without and with plasmon excitation, respectively. $B_c$ in (**e**) and (**f**) are defined by the field at which R reaches 50% of the normal state resistance. The measurements are taken with two-terminal configuration, same to Fig. 1. A contact resistance $R_c = 54.5$ Ω has been deducted.

of the cumulative light heating, we perform temperature distribution simulation by COMSOL Multiphysics software based on the experimental details (Supplementary Fig. 13). Assuming all the absorbed photons are converted to heat, the upper bound of the temperature rise is estimated to be ~ 0.06 K for the typical photon flux we used ($4.68 \times 10^{13}$ s⁻¹mm⁻², 532 nm), almost an order of magnitude smaller than the observed change of $T_c$ (-0.4 K). The parabolic dependences of Δ and $I_c$ in pristine NbSe₂ when increasing the bath temperature (see detailed temperature dependences in Supplementary Fig. 14 and similar results in ref. 40), together with the simulated temperature rise linearly proportional to the photon flux (Supplementary Fig. 13b), is in contrast to the observed linear dependence of both Δ and $I_c$ on photon flux (Fig. 2b, c). These findings suggest that the observed plasmon-induced superconductivity modulation cannot be attributed to the sample heating effect caused by light illumination. More observations are also inconsistent with the heating effect: the quenching of superconductivity modulation effect when increasing NbSe₂ thickness to five layers (Fig. 1i) and the immediate switching behavior with alternating plasmon on/off (discussed later in Fig. 3). These experimental results further validate the model assumption that the resonant plasmon excitation redistributes the electrons in NbSe₂, resulting in a suppression of the effective pairing strength.

The plasmon-induced superconductivity modulation is further studied under a perpendicular magnetic field, as illustrated in Fig. 2d. The extracted critical field $B_c$ is summarized in Fig. 2e as a function of photon flux. $B_c$ shifts to lower fields as the photon flux increases, consistent with the aforementioned plasmon-induced suppression of superconductivity. Figure 2f shows the temperature dependencies of $B_c$ without and with plasmon excitation. The linear temperature dependences of $B_c$ are suggestive of 2D superconductors[35,41] and can be explained by the Ginzburg-Landau (GL) expression[42]

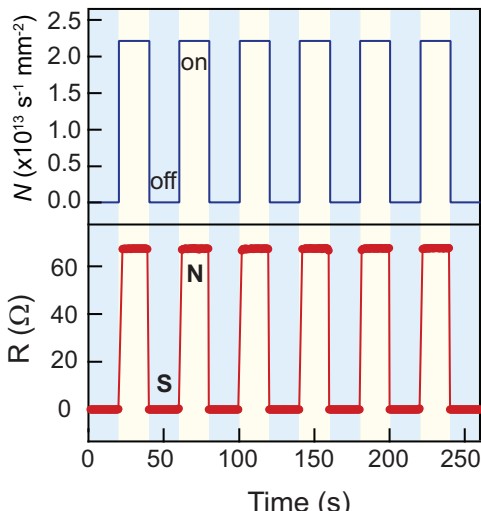

**Fig. 3 | Plasmonic superconducting switch.** Reversible switching between superconducting (S) and normal (N) states of device A1 (AuNPs/hBN/NbSe₂) as the plasmon resonant excitation is alternately turned on and off by the 532-nm light. The superconducting switch operates at a bias current of 28 μA and a temperature of 1.7 K. The measurements are taken with two-terminal configuration, same to Fig. 1. A contact resistance $R_c = 54.5$ Ω has been deducted.

$B_c = \frac{\Phi_0}{2\pi\xi_{GL}^2(0)}(1 - \frac{T}{T_c})$, where $\Phi_0$ and $\xi_{GL}(0)$ are the flux quantum and the zero-temperature in-plane superconducting coherence length, respectively. From the fittings, we can obtain the GL coherence length $\xi_{GL}(0) = 11.8$ nm for the dark case, which is close to the reported values

in intrinsic $NbSe_2$[35,43]. Since the thicknesses of $NbSe_2$ studied in this work are smaller than $\xi_{GL}(0)$, the superconducting phase transition is expected to be of the BKT type, consistent with the scaling law shown in Fig. 2a. From BCS theory, the zero-temperature coherence length is related to the Cooper pair size and is inversely proportional to $T_c$[42]. Under resonant plasmon excitation, the GL coherence length increases to 13.5 nm, consistent with the observed suppression of $T_c$.

### Plasmonic superconducting switch

The observed plasmon-induced superconductivity modulation implies the potential for designing superconducting switch. In stark contrast to previously demonstrated superconducting switch devices that are usually actuated by electrical methods[44–46], the use of light offers distinct advantages of noninvasiveness and ultrafast control. We show the reversible switching between superconducting and normal states in Fig. 3. The detailed operations are as follows: the system is initially at the superconducting states with a high bias current of 28 μA, which is slightly below the critical current of the superconducting-to-normal transition at 1.7 K. Upon turning on plasmon excitation ($N = 2.2 \times 10^{13} \, s^{-1} mm^{-2}$), the critical current is suppressed to a level lower than the bias current, causing an immediate transition to the normal state. Conversely, by turning off plasmon excitation, the critical current exceeds the bias current, resulting in the system returning to the superconducting state. Besides the above switching operation at the high bias current (28 uA, close to $I_c$) and low temperature (1.7 K), we highlight that the superconducting switch can also function at the low bias current (500 nA) and elevated temperature (3.9 K, close to $T_c$), offering a more energy-efficient operation. Such low-bias-current mode switching is shown in Supplementary Fig. 16.

In conclusion, our study demonstrates the modulation of superconductivity through plasmon coupling. The study of light-matter interactions in two-dimensional materials always suffers from low light absorption. By utilizing plasmonic excitation, our work develops an efficient approach to enhance the light-matter coupling within a confined nanometer regime. This approach can find broad applicability in exploring the light-matter interaction in the two-dimensional limit. From a fundamental perspective, the observed plasmon-modulated superconductivity sheds light on the future search for boson-mode-assisted pairing[47,48] or exotic superconductivity mechanisms that go beyond the standard BCS theory. We also expect that the present device can be further combined with Josephson junction geometries[49,50], enabling multifunctional tunability to quantum electronic states. In principle, plasmons can be a generic knob to modulate quantum effect through delicate near-field coupling with other quasiparticles in various geometries and material systems. For instance, the coherent exchange of energy between excitons and plasmons can lead to the condensation of the hybrid polariton states[51,52].

## Methods

### Device fabrication

Before the transfer of $NbSe_2$, hBN and AuNPs, pre-patterned electrodes are fabricated by standard electron-beam lithography, Au/Ti (40 nm / 5 nm) deposition and lift-off process. $NbSe_2$ and hBN (both from HQ Graphene) are mechanically exfoliated onto silicon wafers. Their thicknesses are identified by the optical contrast and confirmed by atomic force microscopy (AFM). We then employ the dry transfer technique to pick up and stack the $NbSe_2$ and hBN onto pre-patterned electrodes. The screening of $NbSe_2$ devices is performed by lateral transport measurements to select appropriate $NbSe_2$ devices for the subsequent transfer of AuNPs. The transfer of AuNPs follows a two-step, two-phase method (Supplementary Note 1). After the transfer of AuNPs, the device is wire-bonded to the transport holder and installed into the cryogenic system (Supplementary Fig. 2).

### Light illumination

Continuous-wave (CW) lasers with wavelengths of 406 nm, 532 nm, 635 nm, 730 nm and 1064 nm are employed in this work. The light is guided into the sample chamber by a fiber and emits out of the fiber ~20 mm away on the top of the device. The light illumination is normal to the sample plane.

### Transport measurements under light

The sample is placed in a helium-free cryostat with the temperature down to 1.7 K and the magnetic field (perpendicular to the sample surface) up to 14 T. All the electrical transport measurements in this study are performed by two-terminal configuration (Fig. 1b) with a current source and a voltage meter. A contact resistance $R_c$ is deducted for simplicity (contact resistance is confirmed by combining $R\text{-}T$ curves and $V\text{–}I$ characteristics). The temperature-dependent and magnetic-field-dependent sample resistances are obtained by standard lock-in technique ($R\text{-}T$ and $R\text{-}B$ curves in Figs. 1d–g and 2d, Supplementary Figs. 3, 4, 5, 11, 15). Lock-in amplifier SR830 is used to provide an AC source of 100 nA with 10 MΩ constant resistance and is also used to probe voltages. The $V\text{–}I$ characteristics and superconducting switch operations are performed with a DC source Keithley 6220 and a nanovoltmeter Keithley 2182 A (Figs. 2a and 3, Supplementary Figs. 10, 12, 14, 16). The alternating plasmon on/off for the superconducting switch operations is controlled by external voltage modulation of 532-nm lasers with a DC voltage source Keithley 2400.

## Data availability

The source data of the main figures in this study have been deposited in the Figshare database under accession code https://doi.org/10.6084/m9.figshare.26083276.v1. Other data supporting the findings of this study are included within the paper and the Supplementary Information file.

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

## Acknowledgements

This work was supported by the National Key Research and Development Program of China (Grant No. 2023YFA1406300), Anhui Provincial Key Research and Development Project (Grant No. 2023z04020008), the Innovation Program for Quantum Science and Technology (Grant No. 2021ZD0302800), the National Natural Science Foundation of China (Grant No. 92165201), the CAS Project for Young Scientists in Basic Research (Grant No. YSBR-046). This work was partially carried out at the USTC Center for Micro and Nanoscale Research and Fabrication. G.C. was also supported by JSPS KAKENHI, WPI-AIMR, Tohoku University, and Anhui Provincial Natural Science Foundation (2208085QA09). W.Q. was supported by the startup foundation.

## Author contributions

C.Z. and G.C. conceived the project. G.C. and Z.W. fabricated the device and performed experiments. M.L. and H.C. carried out the transfer of AuNPs. W.Q., D.W. and Z.Z. provided supporting theoretical analysis. G.C., W.Q., and C.Z. wrote the manuscript with input from co-authors.

## Competing interests

The authors declare no competing interests.
