## [Peer Review File · Nature Communications]

Reversible Modulation of Superconductivity in Thin-Film NbSe₂ via Plasmon CouplingEditorial Note: Parts of this Peer Review File have been redacted as indicated to remove third-party material where no permission to publish could be obtained.

Editorial Note: Figure R9 in this Peer Review File is reproduced with permission from Springer Nature. <Nat. Commun. 14, 2396 (2023)>

Figure R13a in the Peer Review File is reproduced with permission from Springer Nature. <Near-Field Optics and Surface Plasmon Polaritons; *Springer*: Berlin; Vol. 81 (2001)>

REVIEWER COMMENTS

Reviewer #1 (Remarks to the Author):

The article by Guanghui Cheng and coworkers presents experimental data on the quenching of superconductivity in a thin exfoliated film on NbSe₂ exposed to visible radiation. Crucially, modulation of superconductivity only happens when the sample is covered with Au nanoparticles, and the radiation is in resonance with such particles. Modulation of superconductivity is interpreted as arising from the interaction between plasmons in the Au particles and Cooper pairs in the superconductor.

Please note that I am not an expert in optics and plasmons, as my research mainly focuses on transport-related aspects of superconductivity. However, interaction between visible light and superconductivity immediately caught my attention, as this topic is rarely discussed. I find the thickness dependence to be very telling. Exponential dependence on nm scales clearly suggests an effect linked to the evanescent field of EM radiation, not to heating.

I find the paper pleasant to read and interesting. The experimental observations are sound and worth reporting. However, I find some of the claim in the paper concerning applicability of this technique exaggerated. I do not see how superconducting switches and optics can find real-life applicability. This is due to the bulky components needed to shine light on superconducting devices, which will make integration challenging. Also, the authors claim high speed capability of their devices, but they do not demonstrate it. On line 192 they claim "immediate transition to the normal state", but Fig. 3 shows only seconds time scales. It is very hard to judge how fast the transition is. I would have expected, at least, operation in the 10-100 MHz regime for a paper claiming device functionality. Also, the need to pass a large bias current (28 μ A) means extremely high dissipation in the resistive state. As shown in the supplementary material, the device does not switch fast enough with lower bias current. I would appreciate either toning down claims related to immediate applications, or provide concrete examples.

In sum, the paper could be improved before acceptance. I also encourage the editors to contact a reviewer more expert on optical phenomena than myself.

More comments:

Line 31: Reference 4 on superconducting transistors has been thoroughly debunked by several groups. I would either exclude it, or move it together with reference 41 and 43 that relate to quasiparticle injection.

Line 66: Why only two wavelengths were used? What happens if a wavelength not at resonance but close to it is used?

Fig1: What is the hump at $T=4.2$ K shown in c and d? Why is this hump not present in Fig. S2? Why is the curve in Fig. S2 so different, with the onset of resistance above 5K?

Fig2a: the authors assume that the voltage drop at the superconducting-resistive transition matches the gap. While this might be true, the authors should state what assumptions were used.

Line 156: "Both Δ and I_c ...". I do not follow the argument. You say that increasing the photon flux results in a linear decrease of Δ , while increasing temperature results in a parabolic decrease of Δ . With this observation, you exclude temperature-related effects. However, it is not clear to me why it is assumed that photon number and temperature are linear. what if temperature increases as N^2 ?

Fig. S9 is confusing to me. If the modulation of superconductivity would really be due to EM field only, why is it taking several seconds to suppress the S state? Instead, heating might better describe the phenomenon. What is the origin of the asymmetric behavior between going from ON to OFF and from OFF to ON?

Reviewer #2 (Remarks to the Author):

G. Cheng et al report NbS₂-based superconductor-gold nanoparticles (AuNPs) devices and study

their response under 532 (1064)-nm light. With the support of some transport measurements, the authors claim that they have demonstrated a reversible plasmonic superconducting switch with properties that have not been demonstrated before, especially around their fundamental aspect. While I appreciate the hard work of the authors and their strategy on the development of theory, nanofabrication and experiment to support their claim, unfortunately, I am not convinced that what has been discussed and demonstrated in the paper, at least in its present form, convey advances and development in the field and my concerns appear below prevents me recommending this work for publication in Nature Communication. I hope the authors will find the comments useful and constructive:

1) The title of the manuscript is 'plasmonic superconducting switch' but the content and experimental/theory do not have anything with the plasmonic properties of the superconductor in the device and all the plasmon activities are on gold nanoparticles and the way the superconductor was driven by the electromagnetic fields produced from light-matter (in this case AuNPs). Therefore, the author may consider revisiting the title.

2) The authors claim that their results don't follow the thermal response (or what is called as sample heating effect by light illumination) of superconductors. In the paper, this is attributed to the redistribution of the electrons in NbS₂ (though there is commonly no electron alone in the superconducting state- Cooper pairs) due to plasmon resonance excitation. This has not been further discussed as to why this will happen if this is not a thermal response. The authors later claim that a model assumption is developed but then the model/theory is discussed solely in the supplementary information part but not in the main text to be formally reviewed.

3) References [3-6] are all solely electronics and do not have anything about light-matter interactions, so it appears that they are irrelevant to the discussion made in the paper.

4) Without AuNPs what would be the results shown in Fig. 1? A superconductor, especially with a thickness less than the penetration depth, in this case, a few nm, exposed by either 532 nm or 1064 nm light will be suppressed even without the requirement of plasmonic resonances of AuNPs.

5) Line 66: "Light with two different wavelengths (532 67 nm and 1064 nm) used in this study is denoted by the arrows". The authors do not discuss what light is this? Cw or pulse? How the device was illuminated and measured? How was the experiment done? Where are the electrical pads on the device? How the transport measurement was done in the device while AuNPs are assembled to get figures 1-3? A cartoon/schematic showing this would be required.

6) Moreover, the authors do not discuss how the resistance of device A1 was measured, two terminals or four terminals? How the current/voltage were applied/measured?

7) I struggle to follow as to what reason the T_c has shifted to lower temperatures for on resonance but not for the off-resonance while both 532 nm (on-resonance) or 1064 nm (off-resonance) light is significantly above the superconducting gap frequency/energy?

8) Line 86: "Notably, the modulation of superconductivity by on-resonant plasmon excitation (on resonance) is much stronger than that under off-resonant plasmon excitation (off-resonance)." Why? A thin superconductor film should be suppressed by visible-NIR light illumination. I am surprised that the authors show Fig. S2 for R-T of the device without AuNPs under 500 nm light illumination but they don't see any change in resistivity at different light fluxes.

9) I appreciate that AuNPs may enhance light-matter interactions but by what amount? How much electric field would these AuNPs generate that modulates superconductivity in the way needed in this work? What is the origin of this modulation and how can this be explained? Thermal or nonthermal? If light with such a high photon energy can't change the superconductivity of a thin film at all, then I would argue that the claimed modulation of superconductivity by the authors has nothing to do with plasmon excitation too.

10) Again line 89: "Typical R-T curves at the same photon flux ($4.68 \times 10^{13} \text{ s}^{-1} \text{ mm}^{-2}$) are shown at the bottom of the mapping images for comparison. With the dark case as the reference, the change of the transition temperature T_c of NbSe₂ is negligible under off-resonant plasmon excitation. In contrast, T_c decreases from 4.1 K (dark case) to 3.7 K under on-resonant plasmon excitation. Why?"

0) Line 106, "As the thickness of NbSe₂ increases, the modulation coefficient drops rapidly towards zero, indicating the quick quenching of the plasmon-induced modulation effect when getting far away from the AuNPs." Why? How the transport was measured? I see the electrodes are on the surface of the device if I get it correct - so the thickness should not play a major role. What is the skin depth of NbS₂? This needs to be carefully explained in the paper.

1) In Line 110, the authors developed a theory and claim that "Therefore, we conclude that the coupling between NbSe₂ and the dramatically enhanced evanescent field induced by the resonant excitation of plasmons drives the electron distribution in NbSe₂ out of thermal equilibrium, resulting in the suppression of superconductivity". How much and how fast is this EM excitation? Why a 522 nm light can't do this itself especially when there are only a few layers of NbS₂?

2) How Figs. 2 a and d were measured, e.g. V-I? it is needed to plot a proper schematic of the

device and show the transport pads and the measurement technique.

3) The Ginzburg–Landau coherence lengths GL 140 of 11.8 and 13.5 nm without and with plasmon excitation. How much field is needed to destroy superconductivity?

4) Line 148, page 6, the authors claim that “This behaviour suggests the presence of Berezinskii-Kosterlitz-Thouless (BKT) transition, which is previously reported in the thermal-driven evolution of 2D superconductivity at finite temperatures...”

But this effect is predicted to happen in materials with large sheet resistivity (larger than a few kOhms) [see reference 33 in the manuscript]. The authors didn’t analyse anything related to sheet resistivity nor explained why only at that particular optical flux such an effect can happen. Furthermore, the IV measurements alone are not usually enough to determine TBKT in 2D superconductors and more data is needed, e.g. temperature induced vortices/antivortices unbinding model would also be required to make a valid statement [see Nanotechnology 31 (2020) 205002].

5) In line 155, the authors discuss that.... “Both Δ and I_c exhibit linear decreases with increasing photon flux, which is in sharp contrast to the parabolic dependences of Δ and I_c in pristine NbSe2 when increasing the bath 158 temperature These findings suggest that the observed plasmon-induced superconductivity modulation cannot be attributed to the sample heating effect caused by light illumination.... These experimental results further validate the model assumption that the resonant plasmon excitation redistributes the electrons in NbSe2, resulting in a suppression of the effective pairing strength”.

Light-induced modulation or suppression of superconductivity will be based on either thermal or nonthermal effects. ... Here, looking at the device without AuNPs suggests that the superconductivity has nothing to do with light illumination. Again, I would ask how the device was illuminated and the transport was measured, is this vertical or horizontal transport direction?

6) Line 166, Page 7, the authors discuss the magnetic field-dependent data and calculate the GL coherence length of 11.8 nm. Later the authors mentioned that the thickness of NbS2 in their devices is less than this value: “...Since the thicknesses of NbSe2 studied in this work are smaller than GL , the superconducting phase transition is expected to be of the BKT type, consistent with the scaling law shown in Fig. 2a..”.

This seems to be not a valid argument as in this case light with such photon energy even without any plasmon excitation would immediately turn the thin superconductor to normal.

7) Page 8, line 200, “ Fig. 3, indicates the rapid operation capabilities of the plasmonic superconducting switch. In principle, the switch relies on the processes of photon-plasmon energy conversion and particle excitation, which typically occur within picosecond timescales 26, 41, 42 .. 43,44”.

The process of photons or plasmons in superconductors is unique and mostly depends on the material and operation frequency, temperature, etc. References 26, 41, 42, 43, and 44 are all irrelevant to this topic and can’t be used to support the claim. I would say that a thin superconductor film biased close to their critical current, and under optical illumination will show what is shown in Fig. 3. So, the advantages of using AuNPs and hBN in a complex fabrication process have not been justified/explained in the paper.

8) In the method section I tried to find how the hybrid device was measured and therefore how Figs 1-3 were produced. In line 222 the authors state that “Transport measurements were carried out in cryostat system to characterize the superconductivity of NbSe2 before the transfer of AuNPs on the device.”. I believe all three figures in the paper and most in the SI are transport measurements some under light illuminations. I am sorry but I am confused here. The authors repeatedly stated that the results shown in the main paper are for AuNPs/hBN/NbSe2. How transport measurement was done for NbS2 itself and then for the hybrid device? E.g. I struggle to understand Fig. S5 was measure from what device configuration? Is that before the transfer to hBN/NbSe2? If so, I believe the absorbance will be different for only AuNPs and when there is a heterostructure of AuNPs/hBN/NbSe2.

Overall, based on the above comments, unfortunately, I can’t recommend this paper for publication.

Reviewer #3 (Remarks to the Author):

In this manuscript, G. Cheng et al. reported their work on reversible plasmonic superconducting switch. Authors claim that the presence of plasmonic gold nanoparticles (AuNPs) enhance the modulation of the critical temperature (T_c) by 40%. They tested the sample at different

wavelengths, i.e. 532 nm (on resonance) and 1064 nm (off resonance), using 3 layers of NbSe₂. This reviewer is confused whether this enhancement in the modulation is due to AuNPs or simply due to the wavelength of light.

1) The smaller wavelength has less energy (power) compared to the larger wavelength photons; the smaller wavelength will have less effect of the T_c of the material. They tried to convince by measurement of a sample (using 4 layers of NbSe₂), supplementary Fig. S2, with no nanoparticles where T_c does not change with increase in light intensity. This is not an apple-to-apple comparison, due to different layers of NbSe₂. Especially when, in the Fig. 1f, one can see the modulation decreases from 1.4×10^{-14} K s²mm² for 3 layers to 0.4×10^{-14} K s²mm² for 4 layers, even in the presence of AuNPs.

2) In Fig 1c the maximum light intensity is 9×10^{13} S⁻¹ mm⁻² (y-axis) for on resonance, while in the Fig 1d the maximum light intensity is 5×10^{13} S⁻¹ mm⁻² (y-axis) for off resonance. This reviewer wonders why do authors not choose same maximum intensity levels for a better comparison?

3) Author chose the flux as quantitative values for light intensity. But as the higher wavelength has more effect on the T_c, compared to lower wavelength, therefore values in power are better measurements. In term of power, the authors used only $1/4$ of light intensity for off resonance compared to on resonance.

In this reviewer's view, this manuscript can't be published in its present form, due to above mentioned issues. These issues need to be resolved and the manuscript may be resubmitted for review.

Responses to the reviewers' reports (Manuscript NCOMMS-23-51244-T)

Responses to Reviewer #1

General Comment: *The article by Guanghui Cheng and coworkers presents experimental data on the quenching of superconductivity in a thin exfoliated film on NbSe₂ exposed to visible radiation. Crucially, modulation of superconductivity only happens when the sample is covered with Au nanoparticles, and the radiation is in resonance with such particles. Modulation of superconductivity is interpreted as arising from the interaction between plasmons in the Au particles and Cooper pairs in the superconductor.*

Please note that I am not an expert in optics and plasmons, as my research mainly focuses on transport-related aspects of superconductivity. However, interaction between visible light and superconductivity immediately caught my attention, as this topic is rarely discussed. I find the thickness dependence to be very telling. Exponential dependence on nm scales clearly suggests an effect linked to the evanescent field of EM radiation, not to heating.

I find the paper pleasant to read and interesting. The experimental observations are sound and worth reporting. However, I find some of the claim in the paper concerning applicability of this technique exaggerated. I do not see how superconducting switches and optics can find real-life applicability. This is due to the bulky components needed to shine light on superconducting devices, which will make integration challenging. Also, the authors claim high speed capability of their devices, but they do not demonstrate it. On line 192 they claim "immediate transition to the normal state", but Fig. 3 shows only seconds time scales. It is very hard to judge how fast the transition is. I would have expected, at least, operation in the 10-100 MHz regime for a paper claiming device functionality. Also, the need to pass a large bias current (28 μ A) means extremely high dissipation in the resistive state. As shown in the supplementary material, the device does not switch fast enough with lower bias current. I would appreciate either toning down claims related to immediate applications, or providing concrete examples.

Response: We appreciate the reviewer's positive evaluation of this work and constructive suggestions on how to further improve it.

In this work, we observed modulation of superconductivity in few-layer NbSe₂ via plasmon coupling and the functionality of superconducting switch. We agree with the reviewer that the claim of high-speed switching should be supported by time-resolved characterization, which is beyond the capability of our electrical measurement setups. In principle, the switching behavior relies on the processes of photon-plasmon energy conversion and particle excitations, which typically occur on an ultrafast timescale [*Phys. Rev. Lett.* 99, 197001 (2007), *Nano Lett.* 21, 13451351 (2021), *Nat. Photonics* 10, 667-670 (2016)]. The switching speed may also be limited by the rising/falling time of the light itself as well as the response of the electrical circuits.

Here we focus on the fundamental aspect, exploring how the plasmon excitation in gold nanoparticles (AuNPs) couples and modulates the superconductivity of few-layer NbSe₂. Therefore, we have deleted the claim of “*high-speed capability*” and toned down the statements related to the application of the superconducting switch (see highlighted parts in the revised manuscript).

We have deleted the contents related to the rapid operation capabilities and the potential applications in the last two paragraphs on page 11 of the main text.

To emphasize the plasmon-induced superconductivity modulation instead of the capability of the superconducting switch, we have revised the title to “Reversible Modulation of Superconductivity in Few-Layer NbSe₂ via Plasmon Coupling”.

Comment 1: *Line 31: Reference 4 on superconducting transistors has been thoroughly debunked by several groups. I would either exclude it, or move it together with references 41 and 43 that relate to quasiparticle injection.*

Response: Following the reviewer’s suggestion, we have removed reference 4, and added another reference related to light-controlled superconducting transistors:

[4] Suda, M. Kato, R. & Yamamoto, H. M. Light-induced superconductivity using a photoactive electric double layer. *Science* **347**, 743-746 (2015).

Comment 2: *Line 66: Why only two wavelengths were used? What happens if a wavelength not at resonance but close to it is used?*

Response: We thank the reviewer for his/her insightful suggestion, which motivates us to perform further experiments with more illumination wavelengths. Figure R1a-e shows the sample resistance R as a function of temperature T of device A1 (the device studied in the main text) under light illumination with different wavelengths, including 406 nm, 532 nm, 635 nm, 730 nm and 1064 nm. The extracted critical temperature T_c versus photon flux N are shown in Fig. R1f-j, where the T_c modulation coefficients for different wavelengths are obtained by linear fittings (dashed blue lines) and summarized in Fig. R1k. Remarkably, **the modulation coefficient roughly follows the absorbance of AuNPs (green curve), reaching maximum close to the resonance peak of plasmon excitation.** These observations further support our claim that the resonant plasmon excitation of AuNPs plays an essential role in modulating the superconductivity of NbSe₂.

Figure R1. Superconductivity modulation of device A1 (AuNPs/hBN/NbSe₂) with different illumination wavelengths. (a-e) Sample resistance R as a function of temperature T under light illumination of different wavelengths. The measurements are taken with a two-terminal configuration, same to Fig. 1. A contact resistance $R_c = 54.5 \Omega$ has been deduced. (f-j) The critical temperature T_c versus photon flux N for light illumination of different wavelengths. The blue points are experimental results obtained from a-e. The dashed lines are linear fitting curves, whose slopes are defined as the modulation coefficients of T_c . (k) modulation coefficient of T_c as a function of illumination wavelength. The five data points are obtained from the fittings in f-j. The green curve denotes the light absorbance of AuNPs.

We have added the results with different illumination wavelengths (Fig. R1k) to Fig. 1b and added discussions on the wavelength dependence to the second paragraph on page 5 of the revised main text.

We have also added Fig. R1 to the Supporting Information as the new Fig. S4.

Comment 3: *Fig1: What is the hump at $T=4.2\text{K}$ shown in c and d? Why is this hump not present in Fig. S2? Why is the curve in Fig. S2 so different, with the onset of resistance above 5K?*

Response: We thank the reviewer for his/her valuable comments. We would like to reply from the following two aspects:

(a) *Fig1: What is the hump at $T=4.2\text{K}$ shown in c and d? Why is this hump not present in Fig. S2?*

As shown by the blue curves in Fig. 1c,d (reproduced below in Fig. R2a,b), besides the primary resistance drop at ~ 4 K corresponding to the superconducting transition mainly studied in this work, an additional resistance drop is observed at ~ 5 K (“hump” feature). This is likely attributed to the **nonuniformity of the NbSe₂ sample A1**, caused by the nanofabrication processes and the transfer of AuNPs. However, no hump is present in previous Fig. S2 (reproduced below in Fig. R2c), possibly due to the more uniform NbSe₂ sample B1 without the transfer of AuNPs. We note that the hump does not affect the conclusion in the main text since our discussion is focusing on the main resistance drop at ~ 4 K.

Figure R2. Superconductivity modulation of (a,b) a trilayer NbSe₂ device A1 with AuNPs (AuNPs/hBN/NbSe₂) and (c) a pristine 4-layer NbSe₂ device B1 without AuNPs (hBN/NbSe₂). a and c are illuminated by the 532-nm light while b is illuminated by the 1064-nm light.

(b) Why is the curve in Fig. S2 so different, with the onset of resistance above 5K?

The onset of resistance above 5 K in previous Fig. S2 (reproduced above in Fig. R2c) is associated with the **higher T_c of the superconducting transition in pristine 4-layer NbSe₂**, compared to that in trilayer NbSe₂ as shown in Fig. R2a,b. In general, T_c of few-layer NbSe₂ decreases upon reducing the thickness due to the weakening of the interlayer Cooper pairing [Nat. Phys. 12, 139–143 (2016)]. To check the thickness dependence of T_c , we have performed further control experiments on a pristine trilayer NbSe₂ device without AuNPs (hBN/NbSe₂), as shown in Fig. R3. We observed a T_c of ~ 4.5 K (T_c is defined as the temperature at 50% of the resistance drop), close to the T_c of the nonuniform trilayer in Fig. R2a,b (~ 4.1 K for the main resistance drop) and lower than the 4-layer case ($T_c \sim 6.0$ K) in Fig. R2c.

Figure R3. Superconductivity modulation of a pristine trilayer NbSe₂ device B2 (BN/NbSe₂) without depositing AuNPs. 532-nm light illumination with photon fluxes of 0, 0.44, 0.87 and $1.31 \times 10^{13} \text{ s}^{-1} \text{ mm}^{-2}$ is

employed. No noticeable modulation of superconductivity is observed. Right panel: the optical image of device B2 with a two-terminal measurement configuration. A contact resistance $R_c = 90.5 \Omega$ has been deducted.

We have added discussions on the hump feature to the second paragraph on page 4 of the revised main text.

We have added Fig. R3 to the Supporting Information as the new Fig. S3.

Comment 4: Fig2a: the authors assume that the voltage drop at the superconducting-resistive transition matches the gap. While this might be true, the authors should state what assumptions were used.

Response: We thank the reviewer for the insightful comment.

The sharp drop/rise of voltage in the voltage-current (V - I) curves on a linear scale in Fig. S10 (reproduced below in Fig. R4a) and on a logarithmic scale in Fig. 2a (reproduced below in Fig. R4b) represent the transition between the normal state and superconducting state. The derivative of the V - I curves gives differential conductance dI/dV as a function of voltage V , showing two symmetric peaks (Fig. R5a), which correspond to the voltage drops in Fig. R4.

Figure R4. Voltage-current (V - I) characteristics of device A1 on (a) linear scale and (b) logarithmic scale.

Figure R5. (a) Differential conductance spectra of NbSe₂ at different photon fluxes. (b) Schematic of the electron transport at the normal metal/superconductor interface. (c) Photon flux dependence of the superconducting gap 2Δ .

Here we consider the electron transport in our device as electron transmitting through a normal metal/superconductor interface (Fig. R5b). We note that the whole voltage drop V occurs at the interface due to that the total contact resistance is deducted (see Methods). For $|eV| < \Delta$ (2Δ is the superconducting gap), the conductance is dominated by Andreev reflection (green arrows in Fig. R5b). For $|eV| > \Delta$, besides the Andreev-reflection conductance, the transmission of normal electrons into the electron and hole branches of the quasiparticles in the superconductor (red arrows in Fig. R5b) contributes to extra conductance. Therefore, the separation between dI/dV peaks can provide an estimation for the size of superconducting gap 2Δ .

For a more quantitative analysis of the dI/dV spectra, we employ the Blonder-Tinkham-Klapwijk (BTK) model [*Supercond. Sci. Technol.* 23, 043001(2010)] which includes both the single quasiparticle transmission and the Andreev reflection with a finite transparency of the interface. The BTK conductance at zero temperature is given as

$$G(E) = \frac{1 + |\tau|^2 + (-1)^{\text{sgn}(E)} |\tau|^2}{|1 + (-1)^{\text{sgn}(E)} \tau|^2},$$

where τ is the transparency of the barrier, $\tau = \sqrt{1 - \Delta^2/E^2}$

Δ is the superconducting gap, and $E = eV + i\Gamma$ with Γ denoting the broadening parameter. The fittings are shown as red curves in Fig. R5a. The extracted superconducting gap 2Δ by BTK fittings and the peak separations as a function of photon flux N are shown in Fig. R5c. The roughly overlapping results suggest that the conductance peak separation (voltage drop in the V - I curve) matches the superconducting gap.

To reflect these analyses, we have added discussions on the differential conductance peaks to the second paragraph on page 8 of the revised main text and a new Note S3 to the Supplementary Information.

We have added Fig. R5b to the Supporting Information as the new Fig. S12b.

We have also added the reference [*Supercond. Sci. Technol.* 23, 043001 (2010)] to the References in the main text and the Supporting Information.

Comment 5: Line 156: "Both Delta and Ic ...". I do not follow the argument. You say that increasing the photon flux results in a linear decrease of Delta, while increasing temperature results in a parabolic decrease of Delta. With this observation, you exclude temperature-related effects. However, it is not clear to me why it is assumed that photon number and temperature are linear. what if temperature increases as N^2 ?

Response: We thank the reviewer for the valuable comment.

As pointed out by the reviewer, the original manuscript implicitly assumes a linear relation between the incident photon number and the temperature rise of the device. This assumption is

based on the thermal equilibrium system with input from the incident photon flux and output from the dissipation to the cool environment. Considering the heat transfer rate is proportional to the gradient in temperature (Fourier’s law of thermal conduction), the temperature rise is linearly dependent on the photon flux.

To further validate this assumption, we give a simulation of the light-induced heating effect by COMSOL Multiphysics software, as shown in Fig. R6a. The simulation is based on experimental details including heat input by the light illumination and heat output by dissipation to other components and ultimately, to the helium atmosphere. The simulated temperature rise indeed shows a linear dependence on the photon flux, as shown in Fig. R6b.

Figure R6. (a) Temperature distribution simulation by COMSOL Multiphysics software based on the experimental details. Illumination photon flux is $4.68 \times 10^{13} \text{ s}^{-1} \text{ mm}^{-2}$. Assuming all the absorbed photon energy is converted to heat, the temperature at the NbSe₂ plane increases from the based temperature of 2.2 K to 2.26 K under light illumination. (b) The simulated temperature rise ΔT as a function of photon flux N .

We have added discussions on the relation between temperature rise and photon flux to the second paragraph on page 8 of the revised main text.

We have also added Fig. R6 to the Supplementary Information as the new Fig. S13.

Comment 6: *Fig. S9 is confusing to me. If the modulation of superconductivity would really be due to EM field only, why is it taking several seconds to suppress the S state? Instead, heating might better describe the phenomenon. What is the origin of the asymmetric behavior between going from ON to OFF and from OFF to ON?*

Response: We thank the reviewer for his/her insightful comments.

As pointed out by the reviewer, in contrast to the fast switching behavior operating at low temperatures (Fig. 3, reproduced below in Fig. R7a), the superconducting switching close to the critical temperature T_c exhibits a slower transition on the timescale of seconds (previous Fig. S9, reproduced below in Fig. R7b). To elucidate the difference between the two cases, we would like to respond from the following two aspects:

Figure R7. Switching between superconducting (S) and normal (N) states as the plasmon resonant excitation is alternately turned on and off by the 532-nm light. **(a)** Switching operates at a bias current of 28 μA and a temperature of 1.7 K. **(b)** Switching operates at a low bias current of 500 nA and a temperature of 3.9 K close to the critical temperature T_c .

At low temperatures (deep inside the superconducting state), the switching behavior depicted in Fig. R7a is primarily attributed to the coupling between the plasmon-induced evanescent field and electrons in NbSe₂. The associated electron-hole excitations and phonon population with energy surpassing the superconducting gap contribute to Cooper pair breaking, suppressing the superconductivity in a very fast time scale. In this case, the light heating effect (a slow process) does not play a significant role because the system is deeply inside the superconducting state. The small temperature rise (~ 0.06 K for a typical phonon flux of $4.68 \times 10^{13} \text{ s}^{-1} \text{ mm}^{-2}$ as shown in Fig. R6) will not significantly affect the condensation of Cooper pairs, thereby preserving the zero resistance (green shaded area in Fig. R8).

Figure R8. Temperature dependence of the resistance obtained in device A1 (AuNPs/hBN/NbSe₂). No light illumination is applied.

At temperatures close to T_c , the switching behavior depicted in Fig. R7b is contributed by both the fast process of plasmon-related mechanism and the slow process of light heating effect. Specifically, when the temperature is close to the sharp superconducting drop around T_c (red shaded area in Fig. R8), a small temperature rise (~ 0.06 K for typical photon flux) from the light heating effect can lead to a significant change of the sample resistance. Since the light heating effect commonly involves a gradual heat/energy accumulation process, it may contribute to the slow switching behavior observed at 3.9 K in Fig. R7b, on top of the fast plasmon-related process.

As further pointed out by the reviewer, the superconducting switching observed at 3.9 K displays an asymmetric behavior between OFF-to-ON and ON-to-OFF operations (see Fig. R7b). As mentioned above, the slow OFF-to-ON switching is attributed to the gradual heat accumulation process. In contrast, the ON-to-OFF switching is relatively faster possibly because the accumulated energy can rapidly dissipate into the low-temperature environmental bath. This asymmetric behavior between energy accumulation and dissipation processes is also reported in other superconductor-based devices. For example, the current direction-induced superconducting-normal state transitions in twisted bilayer graphene [*Nat. Commun.* 14, 2396 (2023)] depicted in Fig. R9.

Figure R9. Switching between superconducting and normal state in twisted bilayer graphene, adopted from *Nat. Commun.* 14, 2396 (2023).

We have added discussions on the fast/slow and asymmetric features of the switching in the caption of Fig. S16 in the Supplementary Information.

Response to Reviewer#2

General Comment: *G. Cheng et al report NbSe₂-based superconductor-gold nanoparticles (AuNPs) devices and study their response under 532 (1064)-nm light. With the support of some transport measurements, the authors claim that they have demonstrated a reversible plasmonic superconducting switch with properties that have not been demonstrated before, especially around their fundamental aspect.*

While I appreciate the hard work of the authors and their strategy on the development of theory, nanofabrication and experiment to support their claim, unfortunately, I am not convinced that what has been discussed and demonstrated in the paper, at least in its present form, convey advances and development in the field and my concerns appear below prevents me recommending this work for publication in Nature Communication. I hope the authors will find the comments useful and constructive:

Response: We thank the reviewer for his/her careful and insightful review of the manuscript. Below we have addressed all the comments raised by the reviewer in detail and have revised the manuscript accordingly.

Comment 1: *1) The title of the manuscript is ‘plasmonic superconducting switch’ but the content and experimental/theory do not have anything with the plasmonic properties of the superconductor in the device and all the plasmon activities are on gold nanoparticles and the way the superconductor was driven by the electromagnetic fields produced from light-matter (in this case AuNPs). Therefore, the author may consider revisiting the title.*

Response: We thank the reviewer for this suggestion.

As pointed out by the reviewer, our work does not involve the intrinsic plasmon of the superconductor. Instead, we study how the plasmon excitation in gold nanoparticles couples and modulates the superconductivity of few-layer NbSe₂. To avoid confusion, we have changed the title of the manuscript to “Reversible Modulation of Superconductivity in Few-Layer NbSe₂ via Plasmon Coupling”.

Comment 2: *2) The authors claim that their results don’t follow the thermal response (or what is called as sample heating effect by light illumination) of superconductors. In the paper, this is attributed to the redistribution of the electrons in NbSe₂ (though there is commonly no electron alone in the superconducting state- Cooper pairs) due to plasmon resonance excitation. This has not been further discussed as to why this will happen if this is not a thermal response. The authors later claim that a model assumption is developed but then the model/theory is discussed solely in the supplementary information part but not in the main text to be formally reviewed.*

Response: We appreciate the reviewer’s insightful comments, which help us to promote the understanding of the observations. We would like to reply point-by-point as follows:

(a) *The authors claim that their results don’t follow the thermal response (or what is called as sample heating effect by light illumination) of superconductors. In the paper, this is attributed to the redistribution of the electrons in NbSe2 (though there is commonly no electron alone in the superconducting state- Cooper pairs) due to plasmon resonance excitation. This has not been further discussed as to why this will happen if this is not a thermal response.*

Exclusion of thermal or light heating effect

In the original manuscript, we claim that the modulation of superconductivity is not induced by thermal or light heating effect based on the following experimental observations: (1) both Δ and I_c exhibit linear decrease with increasing photon flux (Fig. 2b,c in the main text). This is in contrast to the parabolic dependences of Δ and I_c on the bath temperature (temperature rise is supposed to be linearly dependent on the photon flux, see the simulation in Fig. R10b) observed in pristine NbSe2 (Fig. S14 in the Supplementary Information); (2) The superconductivity modulation coefficient decays exponentially upon increasing the thickness of NbSe2 within the nanometer regime (Fig. 1f in the main text). Such a nanometer characteristic length is much shorter than the typical thermal penetration depth in microns [*J. Appl. Phys.* 121, 175107 (2017)]; (3) the rapid switching between superconducting and normal states with alternating plasmon on and off (see Fig. 3 in the main text). While the heat accumulation usually needs time of seconds to reach thermal equilibrium states. In addition, our earlier work [*Phys. Rev. Lett.* 119, 156803 (2017)] with a similar structure (AuNPs/Al2O3/graphene) reports an enhancement of the quantum coherence of electrons in graphene under plasmon excitation of AuNPs. This also implies that the thermal effect, which typically destroys the quantum coherence, does not play a significant role.

To further support this claim, here we give a quantitative estimation of the temperature rise caused by cumulative light heating by performing simulations using COMSOL Multiphysics software with experimentally realistic inputs (see Fig. R10). By assuming that all the absorbed photon energy is converted to heat, we can obtain the upper bound of the temperature rise of the NbSe2 device, i.e., $\Delta T \sim 0.06$ K for the typical photon flux ($4.68 \times 10^{13} \text{ s}^{-1} \text{ mm}^{-2}$, 532 nm). Such a tiny temperature rise cannot explain the observed significant modulation of T_c of ~ 0.4 K at the same photon flux (Fig. R12a).

Figure R10. (a) Temperature distribution simulation by COMSOL Multiphysics software based on the experimental details. The photon flux of the 532-nm illumination is $4.68 \times 10^{13} \text{ s}^{-1} \text{ mm}^{-2}$. Assuming all the absorbed photon energy is converted to heat, a maximum temperature rise of 2.2 K-to-2.26 K is obtained at the sample plane. (b) The simulated temperature rise ΔT as a function of photon flux N .

Overall, these findings suggest that the observed plasmon-induced superconductivity modulation in the present system at low temperatures is not a thermal response.

Microscopic processes of the plasmon-enhanced superconductivity modulation

We agree with the reviewer that, in the superconducting state, there is no electron alone at the Fermi level due to the condensation of Cooper pairs. To make it clearer regarding the plasmon-induced electron redistribution, here we briefly describe the microscopic processes involved in the plasmon-enhanced light-matter coupling. The excitation of plasmon in AuNPs is associated with an evanescent field that decays exponentially within nanometers [*J. Phys. Chem. C* 123, 1183311839 (2019)]. We first estimate the strength of the evanescent field near the NbSe₂ layer by Finite Difference Time Domain (FDTD) simulations. As shown in Fig. R11b, the on-resonance plasmon excitation of the 4-nm-diameter AuNPs can generate a pronounced evanescent field close to the surface. However, this field decays rapidly when getting far away from the surface of AuNPs. Close to the NbSe₂ plane ($z = -6.5 \text{ nm}$ underneath the AuNPs), the evanescent field strength (Fig. R11c) is at the same level as the far field of propagation light ($(E/E_0)^2 \sim 0.71$, where E_0 is the field of the propagation light), i.e., no significant enhancement of the field strength. This cannot explain the disparity between the substantial and negligible modulations of superconductivity with and without AuNPs (see Fig. R12a,b). However, we emphasize that **the key factor for the plasmon-enhanced light-matter interaction in our work is the higher efficiency of the evanescent field-electron coupling than the propagation light-electron coupling**, as explained below.

Figure R11. FDTD simulation results. (a) The device configuration for FDTD simulation. Note that we use a gold film to replace NbSe₂ layers to simplify the issue involving the optical properties of superconductors. (b) The electric field strength distribution $(E/E_0)^2$ along the xz plane for the section of typical gold nanoparticle. E_0 denotes the far field of the propagation light. (c,d) The electric field strength distribution $(E/E_0)^2$ along the xy plane close to the NbSe₂ layers (detection is set at $z = -6.5 \text{ nm}$) for on resonance and off resonance cases. The blue areas in c and d correspond to the positions directly below the gold nanoparticles. The red areas correspond to the positions directly below the gaps of the nanoparticles.

Figure R12. Modulation of superconductivity of trilayer NbSe₂ devices (a) with and (b) without AuNPs. The 532-nm photon flux used in a is $4.68 \times 10^{13} \text{ s}^{-1} \text{ mm}^{-2}$. The 532-nm photon fluxes used in b are labeled in the legend.

The interaction between the evanescent field and electrons differs significantly from that involving propagation light and electrons [Nat. Rev. Phys. 2, 538–561 (2020)]. Essentially, the wavevector of propagation light with a given frequency is uniquely determined by its dispersion relation (the solid line in Fig. R13a). In contrast, **the plasmon-induced evanescent field is associated with a larger and broader momentum distribution** (the shaded area in Fig. R13a). Typically, the wavevector of a plasmonic nanoparticle exhibits a “bell-shaped” distribution peaking around $1/d$, where d denotes the size of the metallic nanoparticle (see Fig. R13b). By invoking the energy-momentum conservation law in electromagnetic field-electron coupling, the plasmon-induced evanescent field can therefore **activate more electron-hole excitation channels in NbSe₂ than the propagation light**. In short, the excitation of plasmons in AuNPs transforms the propagation light into an evanescent field, enhancing the efficiency of its coupling with electrons in NbSe₂. Therefore, we emphasize again that the plasmon-enhanced light-matter interaction in this work is not due to the enhancement of the field strength, but attributed to the efficient coupling between the plasmon-induced evanescent field and the electrons, which leads to the significant superconductivity modulation.

Figure R13. (a) The $\omega - k_x$ dispersion of the propagation light and surface plasmon polaritons, adopted from *Near-Field Optics and Surface Plasmon Polaritons; Springer: Berlin, 2001; Vol. 81*. (b) The momentum dependence of near-field coupling for a metallic structure (radius $\sim 30 \text{ nm}$), adopted from *Nano Lett. 11, 4701 (2011)*.

Understanding to the plasmon-induced electron redistribution

Under the context discussed above, the evanescent field-induced electron-hole excitations with energies surpassing the superconducting gap contribute to the Cooper pair breaking, entailing the redistribution of electrons above the superconducting gap. From another perspective, we consider a scenario where the system is initialized at the normal state at high temperatures. When the plasmon-induced evanescent field is applied, the evanescent field-induced electron-hole excitations rapidly relax energies via electron-electron and electron-phonon scatterings [*Phys. Rev. X* 7, 041013 (2017); *Nat. Mater.* 17, 586-591 (2018); *Phys. Rev. Lett.* 121, 267001 (2018)], redistributing electrons by depleting their population around the Fermi level, leading to the suppression of the Fermi-level density of states. Consequently, such electron redistribution around the Fermi level at normal state is expected to lower the superconducting transition temperature when cooling down the system. As explained in Note S2 of the Supporting Information, our phenomenological model calculates the superconducting critical temperature by solving the linearized gap equation, wherein the normal-state electron redistribution is incorporated. Within the assumption of the layer-dependent redistribution of electrons in NbSe₂, our theory can well reproduce the nearly-linear dependence of T_c modulation coefficient on the photon flux (Fig. S9a in the Supplementary Information).

Additionally, to provide a crosscheck for the simulated evanescent field, we also conduct FDTD simulations for off-resonant excitation of plasmon. As shown in Fig. R11d, the strength of the evanescent field $(E/E_0)^2 \sim 0.095$, is much smaller than that of the on resonance case (Fig. R11c). This result is consistent with our experimental observation that the superconductivity modulation coefficient for the off resonance case is much smaller than that for on resonance case (modulation coefficients of -1.38×10^{-14} K μm^2 and -0.16×10^{-14} K μm^2 for on- and off-resonant plasmon excitations, respectively, obtained from Fig. 1e in the main text).

The above-mentioned plasmon-related mechanism is further supported by our observations, including: (1) T_c modulation coefficient follows the absorbance of AuNPs and reaches maximum close to the plasmon resonance peak (see further experiments in Fig. R14 and the summarized results in Fig. R14k); (2) the nanometer-scale thickness dependence of superconducting modulation coefficient is consistent with the plasmon-induced evanescent field that decays exponentially within nanometers (Fig. 1f in the main text).

Overall, we would like to emphasize that the key finding of this work is the experimental observation of plasmon-modulated superconductivity, which suggests a new plasmonic knob for modulating superconducting states. We further proposed a possible scenario associated with the plasmon-induced evanescent field, which may play a critical role in enhancing light-matter interactions in superconducting NbSe₂.

Figure R14. Superconductivity modulation of device A1 (AuNPs/hBN/NbSe₂) with different illumination wavelengths. (a-e) Sample resistance R as a function of temperature T under light illumination of different wavelengths. The measurements are taken with a two-terminal configuration, same to Fig. 1. A contact resistance $R_c = 54.5 \Omega$ has been deducted. (f-j) The critical temperature T_c versus photon flux N for light illumination of different wavelengths. The blue points are experimental results obtained from a-e. The dashed lines are linear fitting curves, whose slopes are defined as the modulation coefficients of T_c . (k) modulation coefficient of T_c as a function of illumination wavelength. The five data points are obtained from the fittings in f-j. The green curve denotes the light absorbance of AuNPs.

(b) The authors later claim that a model assumption is developed but then the model/theory is discussed solely in the supplementary information part but not in the main text to be formally reviewed.

Following the reviewer’s suggestion, we have added more descriptions of the theoretical model to the 4th paragraph on page 5 in the revised main text and have also revised Note S2 of the Supporting Information accordingly.

We have added the discussions on the COMSOL simulation to the second paragraph on page 8 of the revised main text.

We have added Fig. R14k to the revised main text as Fig. 1b.

We have added Figs. R10, R11, R14, R15b,c to the Supplementary Information as the new Figs. S13, S7, S4, S3.

Comment 3: 3) References [3-6] are all solely electronics and do not have anything about light-matter interactions, so it appears that they are irrelevant to the discussion made in the paper.

Response: We thank the reviewer for pointing out the improper references.

In the revised main text, we have removed references [3-6] and added the following references related to the light-controlled superconducting devices: “...including cryogenic switches³, superconducting transistors⁴ and tunable qubits⁵”.

Newly added references:

[3] Yang, M. Yan, C. Ma, Y. Li, L. & Cen, C. Light induced non-volatile switching of superconductivity in single layer FeSe on SrTiO₃ substrate. *Nat. Commun.* **10**, 85 (2019).

[4] Suda, M. Kato, R. & Yamamoto, H. M. Light-induced superconductivity using a photoactive electric double layer. *Science* **347**, 743-746 (2015).

[5] Sahu, R. et al. Entangling microwaves with light. *Science* **380**, 718-721 (2023).

Comment 4: 4) Without AuNPs what would be the results shown in Fig. 1? A superconductor, especially with a thickness less than the penetration depth, in this case, a few nm, exposed by either 532 nm or 1064 nm light will be suppressed even without the requirement of plasmonic resonances of AuNPs.

Response: We thank the reviewer for these valuable comments.

We acknowledge that, in principle, superconductivity can be suppressed by light if (1) photon energy is larger than the superconducting gap, and (2) there's strong optical intensity and a high ratio of absorption. The first requirement enables effective particle excitations by the photons and the second two requirements allow for sufficient light-matter interactions.

As reported in earlier studies [*Science* 376, 860-864 (2022), *Phys. Rev. Lett.* 121, 267001 (2018)], ultrafast laser pulses with extremely high intensity have been utilized to investigate the suppression of superconductivity in bulk superconductors with substantial absorption cross sections. The absorbed photons can excite electrons (larger than superconducting gap), relax energy to the phonon reservoir, and then the hot phonons destroy the condensate on an ultrafast time scale [*Phys. Rev. Lett.* 95, 147002 (2005)].

However, the light employed in this work does not meet the two essential conditions of high intensity and substantial absorption, and there's no enough phonon population to quench the superconductivity. Specifically, the intensity of the typical light employed in this work is $\sim 1 \text{ mW/cm}^2$, which is $10^{10} \sim 10^{12}$ times weaker than the typical intensity of laser pulses used in earlier studies (the typical pulse fluence of $50 \mu\text{J/cm}^2$ with a pulse width of 100 femtoseconds in Ref.

Science 376, 860-864 (2022) corresponds to a transient light intensity of 5×10^{11} mW/cm². Furthermore, the few-layer NbSe₂ exhibits much weaker absorbance (<7% for trilayer NbSe₂ based on the absorption coefficient [*J. Appl. Phys.* 41, 4642–4649 (1970)]) than that of the bulk materials. Therefore, it is challenging to directly modulate properties of few-layer NbSe₂ with such a weak light intensity. Indeed, for pristine NbSe₂ devices without AuNPs, no noticeable modulation of superconductivity is observed (see the 4-layer NbSe₂ and trilayer NbSe₂ devices in Fig. R15).

We overcome this challenge by introducing plasmon excitation of AuNPs proximal to few-layer NbSe₂. The resonant excitation of plasmon induces an evanescent field that decays into NbSe₂ and efficiently excites electrons/phonons, leading to the breaking of Cooper pairs in NbSe₂.

Figure R15. Superconductivity transitions of (a) a pristine 4-layer NbSe₂ device B1 and (b) a pristine trilayer device B2 without depositing AuNPs (BN/NbSe₂). Transport measurements are performed by two-terminal measurements with the standard AC lock-in technique. (c) Optical image of device B2 with a two-terminal measurement configuration.

We have added discussions on the light-induced quenching of superconductivity without AuNPs to the third paragraph on page 4 of the revised main text.

We have also added Fig. R15b,c to Supplementary Information as the new Fig. S3.

Comment 5: 5) Line 66: “Light with two different wavelengths (532 nm and 1064 nm) used in this study is denoted by the arrows”. The authors do not discuss what light is this? Cw or pulse? How the device was illuminated and measured? How was the experiment done? Where are the electrical pads on the device? How the transport measurement was done in the device while AuNPs were assembled to get figures 1-3? A cartoon/schematic showing this would be required.

Response: We thank the reviewer for these constructive comments. We would like to reply point-by-point as shown below:

(a) Line 66: “Light with two different wavelengths (532 nm and 1064 nm) used in this study is denoted by the arrows”. The authors do not discuss what light is this? Cw or pulse?

We used continuous-wave (CW) light in this work.

(b) *How the device was illuminated and measured? How was the experiment done?*

As shown in Fig. R16a below, the device is placed in a cryogenic system with a base temperature down to 1.7 K and a magnetic field (perpendicular to the sample surface) up to 14 T. The light is guided into the sample chamber by a fiber and emits out of the fiber ~20 mm away on the top of the device. The light illumination is normal to the sample plane. The transport measurements are performed under continuous light illumination.

Figure R16. Schematics of the transport configuration under light illumination. (a) Transport measurements of a device under light illumination in a cryogenic system. (b) Schematic of the AuNPs/hBN/NbSe₂ device under light illumination. (c) The optical image of the NbSe₂ device A1 (AuNPs/hBN/NbSe₂) with six metallic pads. Two-terminal configuration is illustrated.

(c) *Where are the electrical pads on the device?*

To avoid degradation of NbSe₂ during nanofabrication processes, we fabricated Au/Ti pads on the silicon wafer and then transferred the hBN/NbSe₂ onto the pre-patterned pads. As shown in Fig. R16b,c, the electrical pads are depicted as the yellow bars underneath the NbSe₂ flake.

(d) *How the transport measurement was done in the device while AuNPs were assembled to get figures 1-3? A cartoon/schematic showing this would be required.*

As shown in Fig. R16b, the device is wired to connect to external electrical meters. Note that the wire bonding can penetrate through AuNPs and contact the metallic pads (the AuNPs are not electrically conductive since they are wrapped up by organic molecules). The transport measurements are performed by a two-terminal configuration with a current source and a voltage meter, as illustrated at the bottom of Fig. R16c. The temperature-dependent and magnetic-field-dependent resistances are obtained by standard lock-in technique ($R-T$ and $R-B$ curves in Fig. 1c,d, Fig. 2d). Lock-in amplifier SR830 is used to provide an AC driving current of 100 nA with 10 M Ω constant resistance and also used to probe voltages. The $V-I$ characteristics and superconducting switch operations are performed with a DC source Keithley 6220 and a nanovoltmeter Keithley 2182A (Fig. 2a, Fig. 3). The alternating light on/off for the superconducting switch operations is controlled by external voltage modulation with a DC voltage source Keithley 2400.

We have added the schematics in Fig. R16b,c to Fig. 1a in the revised main text and added Fig. R16a,b to the Supplementary Information as the new Fig. S2.

We have also added transport details in each caption of Fig. 1-3 and enriched the Methods accordingly (highlighted parts in the revised main text).

Comment 6: *6) Moreover, the authors do not discuss how the resistance of device A1 was measured, two terminals or four terminals? How the current/voltage were applied/measured?*

Response: We thank the reviewer for the valuable comment.

The resistance of device A1 is measured by a two-terminal configuration with a current source and a voltage meter, as illustrated at the bottom of Fig. R16c. The temperature-dependent and magnetic-field-dependent sample resistances are obtained by standard lock-in technique (R - T and R - B curves in Fig. 1c,d, Fig. 2d, Fig. S3, Fig. S4, Fig. S5, Fig. S11 and Fig. S15). Lock-in amplifier SR830 is used to provide an AC driving current of 100 nA with 10 M Ω constant resistance and is also used to probe voltages. The V - I characteristics and superconducting switch operations are performed with a DC source Keithley 6220 and a nanovoltmeter Keithley 2182A (Fig. 2a, Fig. 3, Fig. S10, Fig. S12, Fig. S14 and Fig. S16). For the resistances shown in the main text, a contact resistance R_c is deducted for simplicity (contact resistance is confirmed by combining R - T curves and V - I characteristics).

We have enriched the Methods accordingly (highlighted parts in the revised main text).

Comment 7: *7) I struggle to follow as to what reason the T_c has shifted to lower temperatures for on resonance but not for the off-resonance while both 532 nm (on-resonance) or 1064 nm (off-resonance) light is significantly above the superconducting gap frequency/energy?*

Response: We thank the reviewer for the valuable comment.

As explained in detail in the response to Comment 4, in principle, light with photon energies larger than the superconducting gap can suppress superconductivity if there's sufficient optical intensity. However, due to the weak light intensity and low absorbance of the present studied few-layer NbSe₂, the direct light-induced quenching of superconductivity is negligible, verified by the negligible modulation in pristine few-layer NbSe₂ without AuNPs (Fig. R15). In this work, the significant modulation of superconductivity by introducing AuNPs arises from the plasmon-enhanced light-matter interaction.

For **on-resonant excitation of plasmon**, the energy of incident 532 nm light matches the localized surface plasmon resonance (LSPR) of the AuNPs, inducing a significant evanescent field. This evanescent field can penetrate into the NbSe₂ layers, leading to particle excitations in NbSe₂ and breaking Cooper pairs (see the detailed scenario in the response to Comment 2). Therefore, T_c shifts to lower temperatures (Fig. R17a). In contrast, for **off-resonance**, the energy of the incident 1064-nm light is not in resonance with the collective electron oscillation of surface plasmon,

leading to a less pronounced evanescent field. Therefore, no significant change of T_c is observed for 1064-nm light illumination (Fig. R17b).

We perform FDTD simulation to estimate the strength of the evanescent field, as shown in Fig. R11c,d. Indeed, the evanescent field strength for on resonance case is much stronger than that for off resonance case ($(E/E_0)^2 \sim 0.71$ and 0.095 , respectively), consistent with the disparity of the observed superconductivity modulation for the two cases (Fig. R17).

Figure R17. Sample resistance R of device A1 (AuNPs/hBN/NbSe₂) as functions of temperature T under (a) 532-nm light (b) 1064-nm light illumination with the same photon flux ($4.68 \times 10^{13} \text{ s}^{-1} \text{ mm}^{-2}$).

To demonstrate the correlation of T_c modulation with the plasmon resonance of AuNPs, we performed further measurements using different illumination wavelengths (406 nm, 532 nm, 635nm, 730 nm, 1064 nm). The modulation of superconductivity is shown in Fig. R14a-j and the T_c modulation coefficients are summarized in Fig. R14k. Remarkably, **the modulation coefficient follows the absorbance of AuNPs (green curve), reaching maximum at the resonance peak of plasmon excitation.** These observations suggest that the resonant plasmon excitation in AuNPs is essential for modulating the superconductivity of NbSe₂.

We have added descriptions of the plasmon-related scenario to the 4th paragraph on page 5 in the revised main text and have also revised Note S2 of the Supporting Information accordingly.

We have added Fig. R14k to the revised main text as Fig. 1b.

We have added Figs. R15b,c, R14 to the Supplementary Information as the new Figs. S3, S4.

Comment 8: 8) Line 86: “Notably, the modulation of superconductivity by on-resonant plasmon excitation (on resonance) is much stronger than that under off-resonant plasmon excitation (off-resonance).” Why? A thin superconductor film should be suppressed by visible-NIR light illumination. I am surprised that the authors show Fig. S2 for R - T of the device without AuNPs under 532 nm light illumination but they don’t see any change in resistivity at different light fluxes.

Response: We thank the reviewer for these valuable comments.

In principle, as pointed out by the reviewer, the superconductivity of a thin superconductor film can be suppressed by light illumination with energies larger than the superconducting gap (typically by “visible-NIR light” as mentioned by the reviewer). However, as discussed in detail

in the response to Comment 4, this effect requires a high intensity of light (typically using pulsed laser) and substantial absorption cross-section, in order to have a large number of absorbed photons and substantial phonon population to quench superconductivity. In the present study, the weak continuous-wave light and the low absorbance of the pristine few-layer NbSe₂ can hardly meet the above requirements. Indeed, in our control experiment on pristine NbSe₂ devices without AuNPs, no noticeable modulation of superconductivity is observed, as shown in the R - T curves in previous Fig. S2 (reproduced in Fig. R15 with the newly added results of a pristine trilayer NbSe₂ device).

In our work, the introduction of AuNPs can effectively couple the plasmon-induced evanescent field with the electrons of NbSe₂ (see the detailed scenario in the response to Comment 2). Therefore, the on-resonant excitation of plasmon induces a significant evanescent field, which effectively breaks the Cooper pairs and quenches the superconductivity. While the off-resonant excitation of plasmon induces a much less pronounced evanescent field, leading to a weaker modulation of superconductivity. Our further FDTD simulation indeed shows that the strength of the evanescent field for on resonance case is much stronger than that for off resonance case ($(E/E_0)^2 \sim 0.71$ and 0.095 , respectively).

We have added discussions on the light-induced quenching of superconductivity without AuNPs to the third paragraph on page 4 of the revised main text.

We have added Fig. R15b,c to the Supplementary Information as the new Fig. S3.

Comment 9: *9) I appreciate that AuNPs may enhance light-matter interactions but by what amount? How much electric field would these AuNPs generate that modulates superconductivity in the way needed in this work? What is the origin of this modulation and how can this be explained? Thermal or nonthermal? If light with such a high photon energy can't change the superconductivity of a thin film at all, then I would argue that the claimed modulation of superconductivity by the authors has nothing to do with plasmon excitation too.*

Response: We again appreciate the reviewer's valuable comments. We would like to reply point-by-point as shown below:

(a) *I appreciate that AuNPs may enhance light-matter interactions but by what amount? How much electric field would these AuNPs generate that modulates superconductivity in the way needed in this work?*

The extent to which gold nanoparticles enhance light-matter interaction depends on various factors, including the enhancement of absorption, local optical intensity, and the quantum efficiency of the relevant physical processes, such as the plasmon-enhanced fluorescence [*Nat. Photonics* 3, 654–657 (2009)] and the plasmon-enhanced Raman scattering [*Nat. Rev. Phys.* 2, 253–271 (2020)].

Following the reviewer's suggestion, to estimate by what amount AuNPs may enhance light-matter interaction, we estimate the field strength by FDTD simulation. As shown in Fig. R11c, for the on-resonant excitation of the plasmon, the plasmon-induced evanescent field is revealed to be at the same level as that of the imposed laser light, i.e., $(E/E_0)^2 \sim 0.71$. This field amplitude cannot explain

the disparity between the substantial and negligible modulations of superconductivity with and without AuNPs (Fig. R12a,b).

As explained in the response to Comment 2, the key factor responsible for the enhancement of light-matter interaction in this work is that the plasmon-induced evanescent field can couple with electrons in NbSe₂ in a more efficient way than the propagation light, due to the larger and broader momentum distribution of the evanescent field. We note that it's challenging to quantitatively calculate by what amount the AuNPs may enhance the light-matter interaction since it involves the non-equilibrium processes of evanescent field-electron coupling.

To check how much electric field (here referring to plasmon-induced evanescent field) AuNPs generate to modulate superconductivity, we compare the FDTD results for the on-resonant and off-resonant excitation of plasmon. As shown in Fig. R11c,d, the strength of the evanescent field for on resonance case is much stronger than that of the off resonance case, consistent with the disparity of the observed superconductivity modulation coefficients (-1.38×10^{-14} Ks μm^2 and -0.16×10^{-14} Ks μm^2 for on- and off-resonant plasmon excitations, respectively, obtained from Fig. 1e in the main text). To give a typical example, at photon flux of 532-nm light $\sim 4.68 \text{ s}^{-1}\mu\text{m}^{-2}$ ($1.75 \text{ mW}/\text{cm}^2$), the T_c drops from ~ 4.1 K to ~ 3.7 K (Fig. R12a) and the strength of the evanescent field is calculated to be $|E| \sim 67 \text{ V}/\text{m}$ close to the NbSe₂ sample plane.

(b) What is the origin of this modulation and how can this be explained? Thermal or nonthermal?

The origin of this modulation is attributed to the plasmon excitation. As explained in detail in the response to Comment 2, the excitation of plasmons in AuNPs transforms the propagation light into an evanescent field, enhancing the efficiency of its coupling with electrons in NbSe₂. The evanescent field can induce electron-hole excitations and lose energy through electron-electron/phonon scatterings with significant phonon population. Both electron/phonon excitations in NbSe₂ with energies surpassing the superconducting gap can suppress the superconductivity. This scenario is supported by our observations, including illumination wavelength dependence (see Fig. R14k) and thickness dependence (Fig. 1f in the main text), consistent with the character of plasmons in AuNPs.

The superconductivity modulation by plasmon at low temperatures is a **nonthermal effect**. As explained in detail in the response to Comment 2 (*“Exclusion of thermal or light heating effect”*), the simulated upper bound of the temperature rise (~ 0.06 K, Fig. R10) is almost an order of magnitude smaller than the observed modulation of T_c (~ 0.4 K, Fig. R12a) at the same photon flux. Other observations are also inconsistent with the light-induced heating effect, including: (1) the linear dependences of Δ , I_c on photon flux in contrast to the parabolic dependences of Δ , I_c on the bath temperature (temperature rise is supposed to be linearly dependent on the photon flux, see the simulation in Fig. R10b); (2) superconductivity modulation weakens rapidly when increasing thickness to five layers (Fig. 1f in the main text); (3) the fast switching with alternating plasmon on/off (Fig. 3 in the main text).

(c) If light with such a high photon energy can't change the superconductivity of a thin film at all, then I would argue that the claimed modulation of superconductivity by the authors has nothing to do with plasmon excitation too.

We agree with the reviewer that light with high photon energies larger than the superconducting gap can induce particle excitations in superconductors, thereby suppressing the superconductivity. Nevertheless, this effect requires a high light intensity (typically pulsed laser) and substantial absorption cross-section, in order to have sufficient light-matter interaction (see detailed discussion in the response to Comment 4). In the present study, the weak continuous-wave light and the low absorbance of the pristine few-layer NbSe₂ can hardly meet the above requirements. Indeed, in our control experiment on pristine NbSe₂ devices without AuNPs, no noticeable modulation of superconductivity is observed (Fig. R15).

In the present study, the plasmon excitation of AuNPs plays an essential role in the modulation of superconductivity. This is supported by our experimental observations, including: (1) T_c modulation coefficient follows the absorbance of AuNPs and reaches maximum close to the plasmon resonance peak (see Fig. R14k); (2) the rapid quenching of superconductivity modulation effect when increasing NbSe₂ thickness to five layers (Fig. 1f in the main text), consistent with exponential decay of plasmon-induced evanescent field within the nanometer regime. Further discussions on the plasmon-related scenario are in the response to Comment 2.

We have added more discussions on the origin of the modulation to the 4th paragraph on page 5 of the revised main text.

We have also added the discussions on the COMSOL simulation to the second paragraph on page 8 of the revised main text.

Comment 10: *10) Again line 89: “Typical R-T curves at the same photon flux (4.68×10^{13} s⁻¹ mm⁻²) are shown at the bottom of the mapping images for comparison. With the dark case as the reference, the change of the transition temperature T_c of NbSe₂ is negligible under off-resonant plasmon excitation. In contrast, T_c decreases from 4.1 K (dark case) to 3.7 K under on-resonant plasmon excitation. Why?”*

Response: We thank the reviewer for these valuable comments.

In this work, the modulation of superconductivity relies on the plasmon-induced evanescent field, as explained in detail in the response to Comments 2 and 7. The 532 nm light is in resonance with the surface plasmon excitation of AuNPs and induces a significant evanescent field, which effectively couples to electrons in NbSe₂ and quenches the superconductivity (T_c decreases from 4.1K to 3.7 K in Fig. R17a). However, the 1064-nm light is not in resonance with the plasmon of AuNPs, leading to a much less pronounced field. Therefore, no significant change in T_c is observed (Fig. R17b). This is further supported by the FDTD simulation, where the evanescent field strength for the on resonance case is much stronger than that for the off resonance case (Fig. R11c,d), consistent with the disparity of the observed superconductivity modulation for the two cases (Fig. R17).

Comment 11: 11) Line 106, “As the thickness of NbSe₂ increases, the modulation coefficient drops rapidly towards zero, indicating the quick quenching of the plasmon-induced modulation effect when getting far away from the AuNPs.” Why? How the transport was measured? I see the electrodes are on the surface of the device if I get it correct - so the thickness should not play a major role. What is the skin depth of NbSe₂? This needs to be carefully explained in the paper.

Response: We thank the reviewer for his/her valuable comments. We would like to reply point-by-point as shown below:

(a) Line 106, “As the thickness of NbSe₂ increases, the modulation coefficient drops rapidly towards zero, indicating the quick quenching of the plasmon-induced modulation effect when getting far away from the AuNPs.” Why?

This is because the excitation of plasmon in AuNPs results in a significant evanescent field that decays exponentially away from the AuNPs with a characteristic length of nanometers [*J. Phys. Chem. C* 123, 18, 11833–11839 (2019)]. In our scenario, such an evanescent field can penetrate into the NbSe₂ layers placed close to AuNPs, leading to particle excitations and Cooper pair breaking in NbSe₂ (see the detailed scenario in the response to Comment 2). When increasing the thickness, more and more layers of NbSe₂ are placed far away from AuNPs. Therefore, the overall effect is that the plasmon-induced modulation effect in NbSe₂ rapidly quenches. We further propose a theoretical model, where the plasmon-induced evanescent field can drive the layer-dependent redistribution of electrons around the Fermi level of NbSe₂. Model calculations show that the T_c modulation coefficient decays exponentially upon increasing the thickness of NbSe₂, which agrees with the experimental data (Fig. 1f in the main text).

(b) How the transport was measured? I see the electrodes are on the surface of the device if I get it correct - so the thickness should not play a major role.

As explained in detail in the response to Comment 5, the transport is measured by a two-terminal in-plane transport configuration. During the device fabrication, we transferred hBN/NbSe₂ to pre-patterned electrodes, to avoid degradation of NbSe₂ flake during nanofabrication processes. Therefore, the electrodes are in direct contact with the bottom of the NbSe₂ flake (Fig. R16b,c). Since the current is injected from the bottom layer of NbSe₂, the layer thickness indeed plays an essential role, resulting in the exponential decrease of the superconducting modulation coefficient upon increasing the thickness of NbSe₂.

(c) What is the skin depth of NbSe₂? This needs to be carefully explained in the paper.

As pointed out by the reviewer, superconductors generally have a skin effect in the Meissner state, where the transport currents are confined to the surface within London penetration depth [*Proc. Roy. Soc.* 149, 71–88 (1935)]. In the present study, the penetration depth is around 250 nm for NbSe₂ close to zero temperature [*Physica C: Superconductivity* 185–189, 2715–2716 (1991)]. Since our study is focusing on the few-layer NbSe₂ (e.g., trilayer with a thickness of ~2.4 nm), the skin effect does not play a significant role here.

We have added discussions on the skin effect to the third paragraph on page 5 of the revised main text.

Comment 12: 12) In Line 110, the authors developed a theory and claim that “Therefore, we conclude that the coupling between NbSe₂ and the dramatically enhanced evanescent field induced by the resonant excitation of plasmons drives the electron distribution in NbSe₂ out of thermal equilibrium, resulting in the suppression of superconductivity”. How much and how fast is this EM excitation? Why a 532 nm light can’t do this itself especially when there are only a few layers of NbSe₂?

Response: We thank the reviewer for these valuable comments. We would like to reply point-by-point as shown below:

(a) In Line 110, the authors developed a theory and claim that “Therefore, we conclude that the coupling between NbSe₂ and the dramatically enhanced evanescent field induced by the resonant excitation of plasmons drives the electron distribution in NbSe₂ out of thermal equilibrium, resulting in the suppression of superconductivity”. How much and how fast is this EM excitation?

We thank the reviewer’s comments which help us to promote the understanding of the microscopic processes. After further FDTD simulation of the field strength, we found that the field enhancement (factor of ~2) alone cannot explain the disparity of the observed significant and negligible modulation of superconductivity. As explained in the response to Comment 2, we emphasize that the plasmon-induced evanescent field can couple with electrons in NbSe₂ in a more efficient way than the propagation light, due to the larger and broader momentum distribution of the evanescent field. To give a typical example, at photon flux of 532-nm light $\sim 4.68 \text{ s}^{-1}\text{mm}^{-2}$ (1.75 mW/cm^2), the T_c drops from $\sim 4.1 \text{ K}$ to $\sim 3.7 \text{ K}$ (Fig. R12a) and the strength of the evanescent field is calculated to be $|E| \sim 67 \text{ V/m}$ close to the NbSe₂ sample plane.

It’s challenging to give a quantitative estimation of the speed/response time for the plasmon-induced field, which is beyond the ability of our setup and the scope of the present study. We note that the response time relies on the processes of photon-plasmon energy conversion and particle excitations, which typically occur within picosecond timescale based on earlier reports [*Phys. Rev. Lett.* 99, 197001 (2007), *Nano Lett.* 21, 1345-1351 (2021), *Nat. Photonics* 10, 667-670 (2016)].

(b) Why a 532 nm light can’t do this itself especially when there are only a few layers of NbSe₂?

The light-induced quenching of superconductivity is negligible in the studied few-layer NbSe₂ due to the weak light intensity and low absorbance (see the detailed explanation in the response to Comment 4), verified by the negligible modulation in pristine few-layer NbSe₂ without AuNPs (Fig. R15).

Comment 13: 13) How Figs. 2 a and d were measured, e.g. V-I? it is needed to plot a proper schematic of the device and show the transport pads and the measurement technique.

Response: We thank the reviewer for the valuable comment and constructive suggestion.

The transport is measured by a two-terminal configuration with a current source and a voltage meter. The transport configuration and metallic pads are schematically illustrated in Fig. R16. The $V-I$ characteristics in Fig. 2a are performed with a DC source Keithley 6220 and a nanovoltmeter Keithley 2182A. The magnetic-field-dependent resistances in Fig. 2d are obtained by standard lock-in technique. Lock-in amplifier SR830 is used to provide an AC driving current of 100 nA with 10 M Ω constant resistance and is also used to probe voltages.

We have added Fig. R16b,c to Fig. 1 in the revised main text.

We have added Fig. R16a,b to the Supplementary Information as the new Fig. S2.

We have also added transport details in the captions of Fig. 1-3 and enriched the Methods accordingly (highlighted parts in the revised main text).

Comment 14: *14) The Ginzburg–Landau coherence lengths ξ_{GL} of 11.8 and 13.5 nm without and with plasmon excitation. How much field is needed to destroy superconductivity?*

Response: We thank the reviewer for the valuable comment and we would like to reply from the following two aspects:

The zero-temperature Ginzburg-Landau (GL) coherence lengths $\xi_{GL}(T = 0 K)$ characterizes the spatial distance over which the superconducting order parameter varies significantly and has the physical meaning of the size of the Cooper pair. For BCS superconductors, $\xi_{GL}(0) \sim \hbar v_F / T_C$, where v_F denotes the Fermi velocity. With plasmon excitation, the observed increase in ξ_{GL} from 11.8 nm to 13.5 nm (Fig. 2f in the main text) is therefore consistent with the suppression of T_C . At low temperatures when the system is deep inside the superconducting state, the superconducting state is not completely destroyed even with the largest photon flux of the laser (Fig. 1c in the main text).

However, the complete destroy of superconductivity can be observed when parking the system close to the critical transition point. For example, the switching operation close to critical current I_c (Fig. 3 in the main text) shows that the superconductivity can be destroyed and transition to a normal metal state under the plasmon excitation at the photon flux of $2.2 \text{ s}^{-1} \text{ mm}^{-2}$ (0.82 mW/cm^2). The corresponding plasmon-induced evanescent field can be estimated by FDTD to be $|E| \sim 46 \text{ V/m}$. We note that the key factor enhancing the light-matter interaction in this work is not the enhancement of field amplitude, but the more efficient coupling of the evanescent field with the electrons in NbSe₂ compared to the propagation light (as explained in the response to Comment 2).

We have added discussions on the GL coherence length to the second paragraph on page 9 of the revised main text.

Comment 15: *15) Line 148, page 6, the authors claim that “This behaviour suggests the presence of Berezinskii-Kosterlitz-Thouless (BKT) transition, which is previously reported in the thermal-driven evolution of 2D superconductivity at finite temperatures...”*

But this effect is predicted to happen in materials with large sheet resistivity (larger than a few kOhms) [see reference 33 in the manuscript]. The authors didn't analyze anything related to sheet resistivity nor explained why only at that particular optical flux such an effect can happen. Furthermore, the IV measurements alone are not usually enough to determine TBKT in 2D superconductors and more data is needed, e.g. temperature induced vortices/antivortices unbinding model would also be required to make a valid statement [see Nanotechnology 31 (2020) 205002].

Response: We appreciate the reviewer's insightful comments and constructive suggestions.

We agree with the reviewer that the BKT transition typically occurs in materials with large sheet resistance R_{\square} . However, **it may not be necessarily larger than a few k**. For example, the BKT transitions are reported in 7-layer/8-layer Pb films with $R_{\square} \sim 600 \text{ fl} / \sim 200 \text{ fl}$ [*Solid State Communications* 165, 59–63 (2013)], 4-nm-thick Al film with sample resistance $R \sim 90 \text{ fl}$ [*Nanotechnology* 31, 205002 (2020)] and bilayer NbSe₂ with $R_{\square} \sim 70 \text{ fl}$ [*Nat. Phys.* 12, 208–212 (2016)]. In our case, the effective transport area of the trilayer NbSe₂ channel $0.6 \times 3.2 \text{ } \mu\text{m}^2$ and normal-state resistance 88 fl give a normal-state sheet resistance $R_{\square} = 468 \text{ fl}$, which is comparable to sheet resistances of the enumerated atomic thin films that harbor BKT transition.

In the conventional BKT transition, thermal fluctuations lead to the creation of vortices and antivortices. For the zero-resistance phase at low temperatures, bound vortex–antivortex pairs are formed. When temperature increases, the pairs undergo a BKT transition, where they unbind and move freely (a vortex-antivortex “plasma”), inducing a finite resistance. In the present study, by keeping the bath temperature of 2.2 K unchanged while increasing the photon flux, we observe a similar universal scaling relation $V \sim I^3$ at the critical photon flux $N_{\text{BKT}} \sim 6.24 \times 10^{13} \text{ s}^{-1} \text{ mm}^{-2}$ (Fig. 2a in the main text). The underlying mechanism is that the **increase of photon flux enhances the plasmon-related evanescent field and suppresses superconductivity, effectively driving the system to approach the BKT phase transition point.**

We agree with the reviewer that the BKT transition in 2D superconductors should follow the temperature-induced vortices/antivortices unbinding model [*Nanotechnology* 31, 205002 (2020)], which is characterized by the Halperin-Nelson formula [*J. Low Temp. Phys.* 36, 599-616 (1979)] $R(T) = R_0 \exp[-b (\frac{T}{T_{\text{BKT}} - T})^{1/2}]$, where $R(T)$ is the temperature dependence of the resistance, T_{BKT} is BKT transition temperature, R_0 and b are material-specific parameters. As shown in Fig. R18a, the fittings show that the above model provides a reasonable description of the resistance drop during the superconducting transition, further validating the BKT-type transition of the few-layer NbSe₂ in this work. The extracted T_{BKT} at various photon fluxes are summarized in Fig. R18b.

Figure R18. Temperature dependent resistance related to BKT physics. (a) Sample resistance R of device A1 (AuNPs/hBN/NbSe₂) as a function of temperature at different photon fluxes of 532 nm light. The transport measurements are performed with an AC source of 100 nA. The red curves correspond to the fittings using the Halperin-Nelson formula. **(b)** The extracted BKT transition temperature T_{BKT} as a function of photon flux N .

We have added further discussions on the BKT transition to the first paragraph on page 8 of the revised main text.

We have also added Fig. R18 to the Supplementary Information as the new Fig. S11.

Comment 16: 16) In line 155, the authors discuss that.... “Both Δ and I_c exhibit linear decreases with increasing photon flux, which is in sharp contrast to the parabolic dependences of Δ and I_c in pristine NbSe₂ when increasing the bath temperature These findings suggest that the observed plasmon-induced superconductivity modulation cannot be attributed to the sample heating effect caused by light illumination.... These experimental results further validate the model assumption that the resonant plasmon excitation redistributes the electrons in NbSe₂, resulting in a suppression of the effective pairing strength”.

Light-induced modulation or suppression of superconductivity will be based on either thermal or nonthermal effects. ... Here, looking at the device without AuNPs suggests that the superconductivity has nothing to do with light illumination. Again, I would ask how the device was illuminated and the transport was measured, is this vertical or horizontal transport direction?

Response: We thank the reviewer for these constructive comments and we would like to reply in two aspects.

How the device was illuminated and the transport was measured

The device is illuminated by the continuous-wave light and the transport is measured by a two-terminal configuration with a current source and a voltage meter (see the detailed description in the response to Comment 5). As shown in Fig. R16b,c, the transport is measured along the horizontal direction.

The role of light illumination and AuNPs in the superconductivity modulation

The negligible modulation in pristine few-layer NbSe₂ devices without AuNPs (Fig. R15) is because the weak illumination light and the low absorbance of pristine few-layer NbSe₂ do not meet the requirements for direct light-induced modulation in superconductors under strong optical intensities (discussed in detail in the response to Comment 4).

By introducing AuNPs into the device, the excitation of plasmons in AuNPs transforms the propagation light into an evanescent field, enhancing the efficiency of its coupling with electrons in NbSe₂ (as explained in detail in the response to Comment 2). The evanescent field can induce electron-hole excitations and lose energy through electron-electron/phonon scatterings with significant phonon population. Both electron/phonon excitations in NbSe₂ with energies surpassing the superconducting gap can suppress the superconductivity.

Comment 17: 17) Line 166, Page 7, the authors discuss the magnetic field-dependent data and calculate the GL coherence length of 11.8 nm. Later the authors mentioned that the thickness of NbSe₂ in their devices is less than this value: “...Since the thicknesses of NbSe₂ studied in this work are smaller than , the superconducting phase transition is expected to be of the BKT type, consistent with the scaling law shown in Fig. 2a..”.

This seems to be not a valid argument as in this case light with such photon energy even without any plasmon excitation would immediately turn the thin superconductor to normal.

Response: We thank the reviewer for the valuable comment.

We agree with the reviewer that, in principle, the light with photon energies larger than the superconducting gap can suppress the superconductivity without AuNPs. The 532-nm and 1064-nm light used in this work are both far larger than the superconducting gap. However, the light-induced quenching of superconductivity requires a high intensity of light (typically pulsed laser) and substantial absorption cross-section, in order to have sufficient light-matter interaction (see detailed discussions in the response to Comment 4). In the present study, the weak continuous-wave light and the low absorbance of the pristine few-layer NbSe₂ can hardly meet the above requirements. Indeed, in our control experiment on pristine NbSe₂ devices without AuNPs, negligible modulation of superconductivity is observed (Fig. R15).

As mentioned by the reviewer, we obtained the Ginzburg–Landau coherence length of ~11.8 nm by fitting the magnetic field-dependent results (Fig. 2f in the main text). The thickness of the studied trilayer NbSe₂ is ~2.4 nm (0.8 nm per layer), smaller than the Ginzburg–Landau coherence length, consistent with the 2D behavior of the superconductor. From another perspective, the BKT transition is treated as evidence of 2D superconducting transition [*Phys. Rev. Lett.* 50, 1603–1606 (1983)], which is verified in our system by the scaling law in *V-I* curves (Fig. 2a) and temperature dependent resistance (Fig. R18).

We have added discussions on the light-induced quenching of superconductivity without AuNPs to the third paragraph on page 4 of the revised main text.

Comment 18: 18) Page 8, line 200, “Fig. 3, indicates the rapid operation capabilities of the plasmonic superconducting switch. In principle, the switch relies on the processes of photon-plasmon energy conversion and particle excitation, which typically occur within picosecond timescales 26, 41, 42.....43,44”.

The process of photons or plasmons in superconductors is unique and mostly depends on the material and operation frequency, temperature, etc. References 26, 41, 42, 43, and 44 are all irrelevant to this topic and can't be used to support the claim. I would say that a thin superconductor film biased close to their critical current, and under optical illumination will show what is shown in Fig. 3. So, the advantages of using AuNPs and hBN in a complex fabrication process have not been justified/explained in the paper.

Response: We thank the reviewer for the valuable comments and constructive suggestions.

References related to the switching speed

We agree with the reviewer that the time scale of photon/plasmon processes in superconductors is unique and depends on the specific material, operating frequency, temperature, etc. In the original manuscript, we cited the references [26, 41-44] to give a rough speculation of the switching speed which involves photon-plasmon-electron/phonon conversion processes. For example, Ref. 26 [*Phys. Rev. Lett.* 99, 197001 (2007)] reports that the photo-excited electrons in $\text{Bi}_2\text{Sr}_2\text{CaCu}_2\text{O}_{8+\delta}$ can relax to phonons in ~ 100 ps. Nevertheless, we focus on the fundamental aspect, exploring how the plasmon excitation in gold nanoparticles (AuNPs) couples and modulates the superconductivity of the few-layer NbSe₂. Therefore, we have removed those references, deleted the claim of “*high-speed capability*”, and toned down the statements related to the application of the superconducting switch. we have also revised the title to “Reversible Modulation of Superconductivity in Few-Layer NbSe₂ via Plasmon Coupling”.

Advantages of using AuNPs and hBN

The light-induced quenching of superconductivity is negligible in the pristine few-layer NbSe₂ without AuNPs (Fig. R15) due to the weak light intensity and low absorbance (see detailed explanation in the response to Comment 4). The *V-I* characteristics for pristine few-layer NbSe₂ without AuNPs also show negligible modulation under light illumination (Fig. R1 9). In such a case, even if biased close to the critical current, the device without AuNPs will not show switching behavior under light illumination.

Figure R19. Voltage–current (V - I) characteristics of device B3 (hBN/4-layer NbSe₂) on a linear scale. The data are obtained at various photon fluxes of 532-nm light, at a base temperature of 1.6 K.

In contrast, the introduction of plasmon excitation of AuNPs into the NbSe₂ device can significantly enhance the light-matter interaction, leading to the observed modulation of superconducting critical temperature T_c . Both the nanometer-scale thickness dependence (Fig. 1f in the main text) and illumination wavelength dependence (Fig. R14k) of the modulation coefficient are consistent with the characters of plasmon excitation in AuNPs. The superconductivity modulation is also reflected in the change of critical current I_c under light illumination (Fig. 2a in the main text). Therefore, plasmon-induced switching behavior can be observed when parking the system close to I_c (Fig. 3 in the main text).

For the advantage of hBN mentioned by the reviewer, we employ hBN (thickness \sim 5 nm) as a thin insulating layer separating AuNPs and NbSe₂ to avoid charge transfer between them and to protect NbSe₂ from degradation.

We have added discussions on the advantages of AuNPs to the last paragraph (conclusion) of the revised main text.

We have added discussions on the role of hBN to the 4th paragraph on page 2 of the revised main text.

Comment 19: *19) In the method section I tried to find how the hybrid device was measured and therefore how Figs 1-3 were produced. In line 222 the authors state that “Transport measurements were carried out in cryostat system to characterize the superconductivity of NbSe₂ before the transfer of AuNPs on the device.”. I believe all three figures in the paper and most in the SI are transport measurements some under light illuminations. I am sorry but I am confused here. The authors repeatedly stated that the results shown in the main paper are for AuNPs/hBN/NbSe₂. How transport measurement was done for NbSe₂ itself and then for the hybrid device? E.g. I struggle to understand Fig. S5 was measured from what device configuration? Is that before the transfer to hBN/NbSe₂? If so, I believe the absorptance will be different for only AuNPs and when there is a heterostructure of AuNPs/hBN/NbSe₂.*

Response: We thank the reviewer for these valuable comments. We would like to reply point-by-point as shown below:

(a) *In the method section I tried to find how the hybrid device was measured and therefore how Figs 1-3 were produced.*

Following the reviewer’s suggestion, we have added measurement details to each figure caption and enriched the Methods accordingly (highlighted parts in the revised manuscript).

Specifically, the continuous-wave light is guided into the sample chamber and illuminates the device during transport measurements (Fig. R16a,b). The transport measurements are performed

by a two-terminal configuration with a current source and a voltage meter, as illustrated in Fig. R16c. The temperature-dependent and magnetic-field-dependent sample resistances are obtained by standard lock-in technique (R - T and R - B curves in Fig. 1c,d, Fig. 2d). Lock-in amplifier SR830 is used to provide an AC driving current 100 nA with 10 M Ω constant resistance and also used to probe voltages. The V - I characteristics and superconducting switch are performed with a DC source Keithley 6220 and a nanovoltmeter Keithley 2182A (Fig. 2a, Fig. 3). The alternating light on/off for the superconducting switch is controlled by external voltage modulation with a DC voltage source Keithley 2400.

(b) In line 222 the authors state that “...Transport measurements were carried out in cryostat system to characterize the superconductivity of NbSe₂ before the transfer of AuNPs on the device.”. I believe all three figures in the paper and most in the SI are transport measurements some under light illuminations. I am sorry but I am confused here.

The sentence in the Methods cited by the reviewer describes a screening process before the transfer of AuNPs to exclude “bad” devices (either broken or strongly degraded).

All three figures in the main text and those figures in Supplementary Information are from the hybrid device AuNPs/hBN/NbSe₂, except the control experiments on hBN/NbSe₂ (Supplementary Fig. S3). For the transport measurements performed under light illumination (explained in detail in the response to Comment 5), photon fluxes are labeled in the corresponding figures.

(c) The authors repeatedly stated that the results shown in the main paper are for AuNPs/hBN/NbSe₂. How transport measurement was done for NbSe₂ itself and then for the hybrid device? E.g. I struggle to understand Fig. S2 was measured from what device configuration? Is that before the transfer to hBN/NbSe₂? If so, I believe the absorbance will be different for only AuNPs and when there is a heterostructure of AuNPs/hBN/NbSe₂.

The hybrid devices (AuNPs/hBN/NbSe₂) and the pristine devices (hBN/NbSe₂) without AuNPs are fabricated and measured independently. The results shown in the previous Fig. S2 are measured from the device configuration hBN/NbSe₂ without AuNPs (see the device and measurement configurations in Fig. R15). The transport measurements are performed by a two-terminal configuration, as explained in the response to Comment 5.

The reviewer also mentioned that the absorbance is different for only AuNPs and the heterostructure AuNPs/hBN/NbSe₂. We note that the absorption is quite weak for NbSe₂ (< 7% based on the absorption coefficient of NbSe₂) as mentioned in the response to Comment 4. The hBN has negligible absorbance of the visible light due to its large bandgap ~6 eV. However, the absorbance of AuNPs can be up to 27% due to the resonant plasmon excitation (Fig. 11k). Therefore, the absorbance of the heterostructure AuNPs/hBN/NbSe₂ can be roughly approximated by the absorbance of AuNPs. This is reflected in the T_c modulation coefficient of NbSe₂ following the absorbance of AuNPs (Fig. R14k).

Following the reviewer’s suggestion, we have added device details to each figure in the main text and the Supplementary Information, and enriched the Methods accordingly (highlighted parts in the revised manuscript).

Response to Reviewer#3

General Comment: *In this manuscript, G. Cheng et al. reported their work on reversible plasmonic superconducting switch. Authors claim that the presence of plasmonic gold nanoparticles (AuNPs) enhance the modulation of the critical temperature (T_c) by 40%. They tested the sample at different wavelengths, i.e. 532 nm (on resonance) and 1064 nm (off resonance), using 3 layers of NbSe₂. This reviewer is confused whether this enhancement in the modulation is due to AuNPs or simply due to the wavelength of light.*

Response: We thank the reviewer for the insightful comments.

To clarify the origin of the modulation, we would like to first discuss the light-induced suppression of superconductivity without AuNPs. In principle, superconductivity can be suppressed by light with energies larger than the superconducting gap, as reported in earlier studies [*Science* 376, 860864 (2022), *Phys. Rev. Lett.* 121, 267001 (2018)], typically with ultrafast laser pulses with transient high intensity and bulk superconductors with substantial absorption cross sections. The absorbed photon energy transfers to the phonon reservoir and then the hot phonons destruct the condensate on an ultrafast time scale [*Phys. Rev. Lett.* 95, 147002 (2005)]. However, the light employed in this work does not meet the two essential conditions of high optical intensity and substantial absorption. Specifically, the intensity of the continuous-wave light employed in this work is ~ 1 mW/cm², which is $10^{10} \sim 10^{12}$ times weaker than the typical intensity of laser pulses used in earlier studies (the typical pulse fluence of 50 μ J/cm² with a pulse width of 100 femtoseconds in Ref. *Science* 376, 860-864 (2022) corresponds to a transient light intensity of 5×10^{11} mW/cm²). Furthermore, the few-layer NbSe₂ exhibits much weaker absorbance ($< 7\%$ for trilayer NbSe₂ based on the absorption coefficient [*J. Appl. Phys.* 41, 4642–4649 (1970)]) than that of the bulk materials. Indeed, for pristine few-layer NbSe₂ devices without AuNPs, light-induced modulation of superconductivity is negligible (see Fig. R21 below).

We overcome this challenge by introducing plasmon excitation of AuNPs proximal to few-layer NbSe₂. The excitation of plasmons in AuNPs transforms the propagation light into an evanescent field, enhancing the efficiency of its coupling with electrons in NbSe₂. The evanescent field can induce electron-hole excitations and lose energy through electron-electron/phonon scatterings with significant phonon population. Both electron/phonon excitations in NbSe₂ with energies surpassing the superconducting gap can suppress the superconductivity.

To verify the correlation of superconductivity modulation with the plasmon excitation, we perform further experiments with different illumination wavelengths (406 nm, 532nm, 635nm, 730 nm, 1064 nm). The corresponding T_c modulation coefficients are summarized in Fig. R20k. Remarkably, **the modulation coefficient roughly follows the absorbance of AuNPs (green curve in Fig. R20k), reaching maximum at the resonance peak of plasmon excitation.** Therefore, the much stronger superconductivity modulation by 532-nm light compared to 1064-nm light, mentioned by the reviewer, is **not due to its higher energy** (406-nm light has higher energy but weaker modulation). Instead, the resonant plasmon excitation of AuNPs plays an essential role in modulating the superconductivity of NbSe₂.

Figure R20. Superconductivity modulation of device A1 (AuNPs/hBN/NbSe₂) with different illumination wavelengths. (a-e) Sample resistance R as a function of temperature T under light illumination of different wavelengths. The measurements are taken with a two-terminal configuration, same to Fig. 1. A contact resistance $R_c = 54.5 \Omega$ has been deduced. (f-j) The critical temperature T_c versus photon flux N for light illumination of different wavelengths. The blue points are experimental results obtained from a-e. The dashed lines are linear fitting curves, whose slopes are modulation coefficients of T_c . (k) modulation coefficient of T_c as a function of illumination wavelength. The five data points are obtained from the fittings in f-j. The green curve is the absorbance of AuNPs.

More observation follows the character of plasmons in AuNPs: the plasmon-induced quenching of superconductivity depends strongly on the thickness of NbSe₂. As shown in Fig. 1f in the main text, upon increasing the thickness of NbSe₂, the T_c modulation coefficient drops rapidly towards zero. **Such a nanometer-scale thickness dependence is consistent with the plasmon-induced evanescent field, which decays exponentially away from AuNPs within nanometers** [*J. Phys. Chem. C* 123, 18, 11833–11839 (2019)]. Therefore, we provide a possible description of the scenario: the absorbed photons are converted to plasmons of AuNPs, and through coupling with the electrons in NbSe₂, the plasmons lead to electron-hole excitations in NbSe₂. The excited carriers can lose energy through electron-electron/phonon scatterings with a significant phonon population. Both electron/phonon excitations in NbSe₂ with energies surpassing the superconducting gap can eventually suppress the superconductivity.

In the revised main text, we have replaced the previous Fig. 1b by Fig. R20k.

We have also added Fig. R20 to the Supporting Information as the new Fig. S4.

We have added discussions on the light-induced quenching of superconductivity without AuNPs to the third paragraph on page 4 of the revised main text.

Comment 1: 1) The smaller wavelength has less energy (power) compared to the larger wavelength photons; the smaller wavelength will have less effect of the T_c of the material. They tried to convince by measurement of a sample (using 4 layers of NbSe₂), supplementary Fig. S2, with no nanoparticles where T_c does not change with increase in light intensity. This is not an apple-to-apple comparison, due to different layers of NbSe₂. Especially when, in the Fig. 1f, one can see the modulation decreases from $-1.4e-14$ K μm^2 for 3 layers to $-0.4e-14$ K μm^2 for 4 layers, even in the presence of AuNPs.

Response: We thank the reviewer for his/her constructive comments.

To give an “apple-to-apple” comparison, we have performed further control experiments on a trilayer NbSe₂ device, as shown in Fig. R21. No noticeable modulation of superconductivity is observed, indicating that the light-induced superconductivity quenching is negligible in pristine few-layer NbSe₂ without AuNPs. Instead, the observed significant modulation of superconductivity by introducing AuNPs benefits from the plasmon-enhanced light-matter interactions through the evanescent field-electron coupling in NbSe₂.

Figure R21. Superconductivity modulation of a pristine trilayer NbSe₂ device B2 (BN/NbSe₂) without depositing AuNPs. 532-nm light illumination with photon fluxes of 0, 0.44, 0.87 and $1.31 \times 10^{13} \text{ s}^{-1} \text{ mm}^{-2}$ is employed. No noticeable modulation of superconductivity is observed. Right panel: the optical image of device B2 with a two-terminal measurement configuration. A contact resistance $R_c = 90.5 \text{ Ω}$ has been deduced.

As shown in Fig. R20k, The superconductivity modulation does not show a monotonic dependence on the wavelength/energy of illuminating light. Instead, it roughly follows the absorbance of AuNPs (green curve), suggesting the essential role of plasmon excitation in modulating superconductivity.

We have added Fig. R21 to the Supplementary Information as the new Fig. S3.

Comment 2: 2) In Fig1c the maximum light intensity is $\sim 9 \times 10^{13} \text{ s}^{-1} \text{ mm}^{-2}$ (y-axis) for on resonance, while in the Fig1d the maximum light intensity is $\sim 5 \times 10^{13} \text{ s}^{-1} \text{ mm}^{-2}$ (y-axis) for off resonance. This reviewer wonders why do authors not choose same maximum intensity levels for a better comparison?

Response: We appreciate the reviewer’s careful review of this manuscript. The maximum light intensities for on-resonance and off-resonance are $\sim 9 \times 10^{13} \text{ s}^{-1} \text{ mm}^{-2}$ and $\sim 5 \times 10^{13} \text{ s}^{-1} \text{ mm}^{-2}$, respectively. This is due to the different **maximum output powers of the 532-nm/1064-nm lasers**. The typical R - T curves **at the same photon flux** ($4.68 \times 10^{13} \text{ s}^{-1} \text{ mm}^{-2}$) are illustrated in Fig. R22a,b, showing a stronger modulation for the on-resonance case compared to the off-resonance case.

Figure R22. Sample resistance R of device A1 (AuNPs/hBN/NbSe₂) as functions of temperature T under (a) 532-nm light (b) 1064-nm light illumination with the same photon flux ($4.68 \times 10^{13} \text{ s}^{-1} \text{ mm}^{-2}$).

We have added an explanation of the maximum photon fluxes to the second paragraph on page 4 of the revised main text.

Comment 3: 3) Author chose the flux as quantitative values for light intensity. But as the higher wavelength has more effect on the T_c , compared to lower wavelength, therefore values in power are better measurements. In terms of power, the authors used only $1/4$ of light intensity for off resonance compared to on resonance.

Response: We thank the reviewer for his/her comments.

All the illumination light (406 nm ~ 1064 nm) used in this work is much larger than the superconducting gap. As discussed in the response to the General Comment and the observed dependence on the illumination wavelength in Fig. 18k, the modulation of the superconductivity roughly follows the resonant plasmon excitation. Therefore, the modulation cannot be simply attributed to the higher energy of the shorter wavelength light (the 406-nm light has higher energy than 532-nm light but weaker modulation).

We use photon flux as the quantity of light intensity to reflect the scenario: in a quasiparticle picture, the absorbed photons can be converted to plasmons in AuNPs. Through the coupling to electrons in NbSe₂, plasmons can excite electron-hole pairs and relax energy to the phonon population, which contribute to the breaking of Cooper pairs in NbSe₂.

We note that quantifying light intensity in terms of power does not alter our conclusions. The comparison of typical R - T curves for on-/off-resonance cases under the same illumination power of 0.88 mWcm^{-2} is shown in Fig. R23. The superconductivity modulation effect for the on-resonance case is stronger than the off-resonance case. We also replot the T_c modulation coefficient in terms of light power instead of photon flux with the unit changed to Kcm^2/mW . As shown in Fig. R24, the superconductivity modulation still follows the absorbance of AuNPs.

Figure R23. Typical R - T curves of device A1 (AuNPs/hBN/NbSe₂) for dark and on/off resonance cases (at the same illumination power of 0.88 mWcm^{-2}).

Figure R24. T_c modulation coefficient (unit: Kcm^2/mW) as a function of illumination wavelength.

We have added discussions on the light intensity in terms of power to the second paragraph on page 5 of the revised main text.

We have added Figs. R23, R24 to the Supporting Information as the new Figs. S5, S6.

REVIEWER COMMENTS

Reviewer #1 (Remarks to the Author):

I am satisfied by the reply of the authors and their rework of the paper. In particular, I think the authors did a good job in presenting control experiments, which clearly show the importance of 1) having gold nanoparticles, 2) exciting them with the right wavelength. Also given the reply to the other referees, I believe the paper is ready for acceptance.

Reviewer #2 (Remarks to the Author):

The authors revised the manuscript but I believe that the data doesn't support the claim made in the paper, and the paper has now lost the novelty for publication in Nature Communications. It is hard to understand the role of Au nanoparticles (AuNPs) in the modulation of the superconductivity of NbSe₂ in this paper. The authors provided very unclear COMSOL simulations to support their claims but they miss describing the model properly, especially around the superconductive properties of NbSe₂ and in the standard of Nature Communication journal. e.g., the authors do not discuss how the COMOSL simulation was done, how superconductivity was modelled in NbS₂ thin films, how large was the area of the device and under what light illumination (what power) the simulation was performed. Was there substrate considered? How the plasmon excitations of AuNPs and their coupling to NbSe₂ was simulated, etc? Based on the above concerns and that the paper has nothing with the plasmonic properties of NbSe₂ and light-induced quenching of superconductivity is known widely, I am sorry to say that I can't find this paper suitable for publication in Nature Communication.

Further comments for the authors' consideration:

-The authors further claim that their earlier work on graphene '[Phys. Rev. Lett. 119, 156803 (2017)]' suggest that the thermal effect can be excluded. Comparing graphene (as a normal material) with NbSe₂ (as a superconductor) cannot be concluded in the way the authors claim so I cannot agree with the authors about the validity of their statement as NbS₂ and graphene offer two different physics one in Room and the other at low temperatures.

-The authors further show the field magnitude extracted from their COMSOL simulation. Here, they use a gold layer instead of NbSe₂ superconductor to simplify the effect of the superconductor!! This cannot be a valid approach again because gold and superconductors have different light-matter interaction physics. For example, most fields generated by the AuNPs will be screened by the superconductor in the first instance.

-Moreover, the authors claim that "However, we emphasize that the key factor for the plasmon-enhanced light-matter interaction in our work is the higher efficiency of the evanescent field electron coupling than the propagation light-electron coupling".

But there is no evanescent field 'electron' coupling enhancement in the NbSe₂ superconductor due to 1) fast decay of the evanescent field, 2) screening of weak field by the superconductor, 3) there are no free electrons in a superconductor. Moreover, light-matter interactions in superconductors, breaking Cooper-pairs in the superconductor, generating QPs and suppressing the superconductivity, are all known to the community, as an effect that has been demonstrated decades ago.

-The authors show Fig. R12 and claim that what was observed in 'a' is due to the presence of AuNPs but this can't be observed in 'b' as there are no AuNPs.

Showing two curves taken in two completely different laser power conditions cannot be a convincing claim. What is shown is that in 'a', the device is illuminated by a 532-nm light with a photon flux of $4.68 \times 10^{13} \text{ s}^{-1} \text{ mm}^{-2}$ while 'b' shows the results for when the device without AuNPs was illuminated by photon fluxes of significantly weaker intensity (almost a factor of 5 difference between the photon fluxes in a and b). This is inconsistent experimental data which does not support the claim made in the paper.

-The argument in "Microscopic processes of the plasmon-enhanced superconductivity modulation" seems to be irrelevant to light-matter interactions in superconductors. The authors refer to Nat. Rev. Phys. 2, 538–561 (2020) and claim that "...activate more electron-hole excitation channels in

NbSe₂ than the propagation light. In short, the excitation of plasmons in AuNPs transforms the propagation light into an evanescent field, enhancing the efficiency of its coupling with electrons in NbSe₂.”

The Nat. Rev. Phys. 2, 538–561 (2020) paper is about light-matter interactions in normal (non-superconducting materials). But where are the free electrons in superconductor NbS₂ the authors address?

-In page 16, the authors stated “However, the light employed in this work does not meet the two essential conditions of high intensity and substantial absorption, and there’s not enough phonon population to quench the superconductivity. Specifically, the intensity of the typical light employed in this work is $\sim 1 \text{ mW/cm}^2$ ”.

However, the authors don’t discuss how much power/field is needed to modulate superconductivity in NbSe₂ and how much power their AuNPs will generate for a 1 mW power illumination. And what one expect if the laser power is increased?

-How cw light- superconductor interaction is compared to ultrafast pulse light excitations while in the former there is no window for the recovery of broken Cooper pairs so a constant suppression of superconductivity is expected.

-On page 20 of the answer to the referee question file, the authors further discuss that “The modulation of superconductivity is shown in Fig. R14a-j and the T_c modulation coefficients are summarized in Fig. R14k. Remarkably, the modulation coefficient follows the absorbance of AuNPs (green curve), reaching a maximum at the resonance peak of plasmon excitation. These observations suggest that the resonant plasmon excitation in AuNPs is essential for modulating the superconductivity of NbSe₂”.

Here, only 532 has a wide modulation as opposed to other illuminations. While looking at the curve R14K, the maximum absorbance occurs at 580 nm. This contradicts what the authors claim as there are no major modulations observed for 406 nm, 730 nm, 635 nm and 1064 nm, all almost on the absorption peak.

-The authors further discuss that “In the present study, the plasmon excitation of AuNPs plays an essential role in the modulation of superconductivity. This is supported by our experimental observations, including (1) the T_c modulation coefficient follows the absorbance of AuNPs and reaches maximum close to the plasmon resonance peak (see Fig. R14k); (2) the rapid quenching of superconductivity modulation effect when increasing NbSe₂ thickness to five layers (Fig. 1f in the main text),”.

But the modulation of superconductivity demonstrated by the authors is in the time scale of a second and this cannot be seen as a rapid quenching of superconductivity.

-On page 24, the authors agree that their NbSe₂ film cannot be uniformly patterned in the substrate (evidenced by two T_c points in the transport), and later they mention that “the layer thickness indeed plays an essential role”, “Therefore, the electrodes are in direct contact with the bottom of the NbSe₂ flake”.

This suggests that the authors have no control over the thickness of their devices, the number of layers might be more or less on different parts of the devices and so the device behaves differently depending on the illumination spot.

Reviewer #3 (Remarks to the Author):

This reviewer’s main question was whether the suppression of the superconductivity is due to plasmonic effect or the heating effect of the light. The authors answered that question by doing more measurements and updating the Fig.1b, Fig.S3 and adding new figures FigS3, FigS4, and

FigS7. They also added the measurement for the wavelengths of 406 nm, 635 nm, and 730 nm. It is now more convincing to claim that the suppression of the superconductivity is solely due to plasmonic effect, instead of heating from the light. It is because at wavelength of 406 nm there is no suppression of superconductivity compared to plasmonic resonance wavelength of 532 nm, even though 406 nm carries more energy. This reviewer appreciates the authors' hard work. This reviewer thinks this article will be interesting for the readers and should be published in Nature communication.

A few minor suggestions.

1. By making the color scales of FigS7c and FigS7d same, it may be easier visually of readers to see the difference in on and off resonance.
2. Is it necessary to say "Few-Layer NbSe₂" in the title or "thin-film NbSe₂" will also work?

Responses to the reviewers' reports (Manuscript NCOMMS-23-51244A)

Responses to Reviewer #1

General Comment: *I am satisfied by the reply of the authors and their rework of the paper. In particular, I think the authors did a good job in presenting control experiments, which clearly show the importance of 1) having gold nanoparticles, 2) exciting them with the right wavelength. Also given the reply to the other referees, I believe the paper is ready for acceptance.*

Response: We appreciate the reviewer's recognition of our last response and the control experiments, as well as his recommendation for publication.

Responses to Reviewer #2

General Comment: *The authors revised the manuscript but I believe that the data doesn't support the claim made in the paper, and the paper has now lost the novelty for publication in Nature Communications. It is hard to understand the role of Au nanoparticles (AuNPs) in the modulation of the superconductivity of NbSe₂ in this paper. The authors provided very unclear COMSOL simulations to support their claims but they miss describing the model properly, especially around the superconductive properties of NbSe₂ and in the standard of Nature Communication journal. e.g., the authors do not discuss how the COMOSL simulation was done, how superconductivity was modeled in NbSe₂ thin films, how large was the area of the device and under what light illumination (what power) the simulation was performed. Was there substrate considered? How the plasmon excitations of AuNPs and their coupling to NbSe₂ was simulated, etc? Based on the above concerns and that the paper has nothing with the plasmonic properties of NbSe₂ and light-induced quenching of superconductivity is known widely, I am sorry to say that I can't find this paper suitable for publication in Nature Communication.*

Response: We thank the reviewer for his/her valuable comments. We would like to reply from the following two aspects:

The role of AuNPs in the superconductivity modulation of NbSe₂

As explained in our last response, due to the weak illumination light and the low absorbance of pristine few-layer NbSe₂, no noticeable modulation is observed in pristine few-layer NbSe₂ without AuNPs (Fig. S3 in SI). By introducing AuNPs into the device, we observed significant superconductivity modulation. Such modulation is attributed to the plasmon excitation in AuNPs, well supported by the experimental observations, including the wavelength dependence (Fig. 1b in the main text), thickness dependence (Fig. 1f), and control experiments without AuNPs (Fig. S3). We further propose a scenario: the plasmon excitation in AuNPs transforms the light into an evanescent field, enhancing the efficiency of its coupling with electrons in NbSe₂; the evanescent field can induce electron-hole excitations and lose energy through electron-electron/phonon

scatterings with significant phonon population; both electron/phonon excitations in NbSe₂ with energies surpassing the superconducting gap can suppress the superconductivity. Based on the above statements, we emphasize that **the novelty of this work** is the experimental observation of plasmon-modulated superconductivity, which suggests a new and efficient knob for modulating superconducting states in contrast to the light-induced superconductivity modulation.

FDTD simulation

To quantitatively estimate the plasmon-induced evanescent field, we conducted Finite Difference Time Domain (FDTD) simulation of the field distribution (not COMSOL). We note that it is challenging to model the optical properties of superconductors using FDTD and we employ a gold film to mimic the NbSe₂ layers. This is based on the similar electromagnetic responses of superconductors and normal metals when the electromagnetic energy surpasses the superconducting gap (the photon energy in our work is ~ 2.33 eV) [*Supercond. Sci. Technol.* 26 114001(2013)]. The dimensions of the simulation are 7×12.1 nm², including one periodic structure of AuNPs array with the underlying infinite-size trilayer NbSe₂. The light illumination power E_0^2 is unified as 1. To facilitate comparisons with free-space light intensity, the simulation strength of the evanescent field is normalized as $(E/E_0)^2$. The substrate SiO₂ is included in the simulation.

We thank the reviewer for the constructive comments and have added the above details of FDTD simulation to the caption of Fig. S7 in the revised Supplementary Information.

Comment 1: *The authors further claim that their earlier work on graphene [Phys. Rev. Lett. 119, 156803 (2017)] suggest that the thermal effect can be excluded. Comparing graphene (as a normal material) with NbSe₂ (as a superconductor) cannot be concluded in the way the authors claim so I cannot agree with the authors about the validity of their statement as NbSe₂ and graphene offer two different physics one in Room and the other at low temperatures.*

Response: We note that both experiments were performed at low temperatures of several Kelvin. As explained in our last response, we quote our earlier study [*Phys. Rev. Lett.* 119, 156803 (2017)] to give additional evidence that the light-induced heating effect (light absorption in AuNPs and energy dissipation) does not play a significant role in affecting the transport behaviors of a material that is placed a few nanometers away from AuNPs. In our earlier study, if the heating effect is significant, the weak localization effect of graphene should be quenched. However, we observed an enhancement of the weak localization effect, indicating the heating effect is negligible.

Comment 2: *The authors further show the field magnitude extracted from their COMSOL simulation. Here, they use a gold layer instead of NbSe₂ superconductor to simplify the effect of the superconductor!! This cannot be a valid approach again because gold and superconductors have different light-matter interaction physics. For example, most fields generated by the AuNPs will be screened by the superconductor in the first instance.*

Response: As explained in the response to the General Comment, it is challenging to model the optical properties of superconductors using FDTD (not COMSOL). For simplification, we employ a gold film to mimic NbSe₂ layers based on the fact that superconductor and normal metal have similar electromagnetic responses when the electromagnetic energy surpasses the superconducting gap (the photon energy in our work is ~2.33 eV) [*Supercond. Sci. Technol.* 26 114001(2013)]. Therefore, it is reasonable to use gold film to mimic NbSe₂ layers in the simulation.

We have added more details of the FDTD simulation to the caption of Fig. S7 in the revised Supplementary Information.

Comment 3: *Moreover, the authors claim that “However, we emphasize that the key factor for the plasmon-enhanced light-matter interaction in our work is the higher efficiency of the evanescent field electron coupling than the propagation light-electron coupling”.*

But there is no evanescent field ‘electron’ coupling enhancement in the NbSe₂ superconductor due to 1) fast decay of the evanescent field, 2) screening of weak field by the superconductor, 3) there are no free electrons in a superconductor. Moreover, light-matter interactions in superconductors, breaking Cooper-pairs in the superconductor, generating QPs and suppressing the superconductivity, are all known to the community, as an effect that has been demonstrated decades ago.

Response: We thank the reviewer for the constructive comments and we would like to reply point-by-point as follows:

1) The evanescent field indeed decays fast. However, there’s still finite field strength up to tens of nanometers away. For example, scanning near-field optical microscopy (SNOM) uses an evanescent field emitted from the metallic tip to probe the near-field response, which can work at distances up to 30 nm away (see Fig. 15 in [*Near-Field Optics and Surface Plasmon Polaritons*, edited by S. Kawata; Springer: Berlin, 2001; Vol. 81]). Our simulated FDTD results (Fig. S7c) show a finite field strength $(E/E_0)^2 \sim 0.7$ at the NbSe₂ sample plane. Accordingly, we have added more discussions in the revised manuscript.

2) The screening of electric field by the superconductor is believed to be determined by the London penetration depth due to the Meissner effect [*Phys. Rev. B* 69, 214515 (2004)]. The penetration depth is ~250 nm for NbSe₂ close to zero temperature [*Physica C: Superconductivity* 185–189, 2715-2716 (1991)]. Since the thickness of the studied trilayer NbSe₂ is only ~2.4 nm, the field cannot be fully screened by the “two-dimensional” superconductor. Accordingly, we have added more discussions in the revised manuscript.

3) In our last response, we concurred with the reviewer’s comment that there’s no electron distribution at the Fermi level in a superconductor. Instead, electrons combine to form Cooper pairs, which condense into the superconducting state. When the plasmon-induced evanescent field is applied, it can induce electron-hole excitations with energies surpassing the superconducting gap. The photoexcited carriers rapidly relax energies via electron-electron and electron-phonon

scatterings, thereby contributing to the breaking of Cooper pairs, leading to a suppression of superconductivity in NbSe₂.

4) As explained in our last response, light-induced modulation in pristine superconductors requires very strong light intensity, which is generally not possible for continuous-wave light. The plasmon excitation, however, can transform light into the evanescent field and efficiently excite electron/phonon population in NbSe₂. Therefore, we emphasize that the novelty of this work is the experimental observation of plasmon-modulated superconductivity, which suggests a new and efficient knob for modulating superconducting states in contrast to the light-induced modulation.

Comment 4: *The authors show Fig. R12 and claim that what was observed in ‘a’ is due to the presence of AuNPs but this can’t be observed in ‘b’ as there are no AuNPs.*

Showing two curves taken in two completely different laser power conditions cannot be a convincing claim. What is shown is that in ‘a’, the device is illuminated by a 532-nm light with a photon flux of $4.68 \times 10^{13} \text{ s}^{-1} \text{ mm}^{-2}$ while ‘b’ shows the results for when the device without AuNPs was illuminated by photon fluxes of significantly weaker intensity (almost a factor of 5 difference between the photon fluxes in a and b). This is inconsistent experimental data which does not support the claim made in the paper.

Response: We thank the reviewer for the constructive comments.

Following the reviewer’s suggestion, we have added new figures under similar photon fluxes ($1.56 \times 10^{13} \text{ s}^{-1} \text{ mm}^{-2}$ and $1.31 \times 10^{13} \text{ s}^{-1} \text{ mm}^{-2}$ for Fig. R1a and R1b, respectively), suggesting significant superconductivity modulation of NbSe₂ with AuNPs and negligible superconductivity modulation without AuNPs. Accordingly, we have replaced Fig. S3 with Fig. R1.

Figure R1. Modulation of superconductivity of trilayer NbSe₂ devices (a) with and (b) without AuNPs. The 532-nm photon fluxes used in a and b are $1.56 \times 10^{13} \text{ s}^{-1} \text{ mm}^{-2}$ and $1.31 \times 10^{13} \text{ s}^{-1} \text{ mm}^{-2}$, respectively.

Comment 5: *The argument in “Microscopic processes of the plasmon-enhanced superconductivity modulation” seems to be irrelevant to light-matter interactions in superconductors. The authors refer to Nat. Rev. Phys. 2, 538–561 (2020) and claim that “...activate more electron-hole excitation channels in NbSe₂ than the propagation light. In short, the excitation of plasmons in AuNPs*

transforms the propagation light into an evanescent field, enhancing the efficiency of its coupling with electrons in NbSe₂.”

The Nat. Rev. Phys. 2, 538–561 (2020) paper is about light-matter interactions in normal (non-superconducting materials). But where are the free electrons in superconductor NbSe₂ the authors address?

Response: We thank the reviewer for the valuable comments.

The electrons combine into Cooper pairs in the superconducting gap, but free electrons do exist outside the gap. As explained in the response to Comment 3, the plasmon-induced evanescent field, with energies surpassing the superconducting gap, can induce electron-hole excitations across the superconducting gap. The photoexcited carriers rapidly relax energies via electron-electron and electron-phonon scatterings, thereby contributing to the breaking of Cooper pairs, leading to a suppression of superconductivity in NbSe₂.

Comment 6: *In page 16, the authors stated “However, the light employed in this work does not meet the two essential conditions of high intensity and substantial absorption, and there’s not enough phonon population to quench the superconductivity. Specifically, the intensity of the typical light employed in this work is ~ 1 mW/cm²”.*

However, the authors don’t discuss how much power/field is needed to modulate superconductivity in NbSe₂ and how much power their AuNPs will generate for a 1 mW power illumination. And what one expect if the laser power is increased?

Response: Our experiments show negligible superconductivity modulation of pristine NbSe₂ with typical powers of our continuous-wave light (Fig. S3 in SI). This is because the light-induced superconductivity modulation requires a significantly strong light intensity, as explained in our last response.

By employing AuNPs, significant modulation of the superconductivity of NbSe₂ is observed under the typical photon flux of $4.68 \times 10^{13} \text{ s}^{-1} \text{ mm}^{-2}$ (power of 1.75 mW/cm^{-2}) (Fig. 1c,d in the main text). As mentioned in our last response, this power corresponds to an evanescent field strength $|E| \sim 67 \text{ V/m}$. Based on the absorbance of $\sim 27\%$ (Fig. 1b in the main text) of AuNPs, illumination power of 1mW means absorbed power of 0.27 mW by AuNPs. And when increasing laser power, as shown in Fig. 1e in the main text, we observed an almost linear increase in the superconductivity suppression.

Comment 7: *How cw light- superconductor interaction is compared to ultrafast pulse light excitations while in the former there is no window for the recovery of broken Cooper pairs so a constant suppression of superconductivity is expected.*

Response: We thank the reviewer for the valuable comments.

The suppression of superconductivity is determined by the photoexcited particle density $N_e(t)$ (including electrons and phonons)[*Phys. Rev. B* 84, 180507(R) (2011), *Science* 376, 860-864 (2022)]. For the transient pulsed light, the instant photoexcited particle density right after a pulse is $N_e=\alpha F$, where F is the pulse fluence and α is the photon-to-particle conversion efficiency. The typical pulse fluence of $50 \mu\text{J}/\text{cm}^2$ gives rise to a density $N_e=\alpha \times 50 \mu\text{J}/\text{cm}^2$. For the CW light, the dynamical equilibrium state is reached with both the excitation process αP and recovery process $N_e(t)/\tau$, where P is the light power and τ is the characteristic recovery time (typically $\tau \sim 3\text{ps}$ in Ref.

Science 376, 860-864 (2022)). A simple model can be written as $\dot{N}_e = \alpha P - N_e/\tau = 0$. Then the particle density $N_e = \tau \alpha P = \alpha \times 3 \times 10^{-9} \mu\text{J}/\text{cm}^2$ for the typical cw power $P = 1 \text{ mW}/\text{cm}^2$ in our work. Therefore, the photoexcited particle density N_e under CW light, which determines the superconductivity modulation, is $\sim 10^{10}$ times weaker than the pulsed light case.

Comment 8: *On page 20 of the answer to the referee question file, the authors further discuss that “The modulation of superconductivity is shown in Fig. R14a-j and the T_c modulation coefficients are summarized in Fig. R14k. Remarkably, the modulation coefficient follows the absorbance of AuNPs (green curve), reaching a maximum at the resonance peak of plasmon excitation. These observations suggest that the resonant plasmon excitation in AuNPs is essential for modulating the superconductivity of NbSe2”.*

Here, only 532 has a wide modulation as opposed to other illuminations. While looking at the curve R14K, the maximum absorbance occurs at 580 nm. This contradicts what the authors claim as there are no major modulations observed for 406 nm, 730 nm, 635 nm and 1064 nm, all almost on the absorption peak.

Response: We thank the reviewer for the valuable comments.

The wavelength dependence does not contradict but supports our claim. The prominent peak of resonant plasmon excitation in AuNPs is $\sim 567 \text{ nm}$ (Fig. 1b in the main text) with a relatively large peak width of $\sim 200 \text{ nm}$. The 532-nm light we employed is quite close to the resonant plasmon peak. We observed significant superconductivity modulation under the 532 nm illumination (close to the plasmon excitation) and less pronounced modulation under 406 nm, 760 nm, 635 nm and 1034nm (far away from the plasmon excitation), supporting the claim that the superconductivity modulation is attributed to the plasmon excitation.

Comment 9: *The authors further discuss that “In the present study, the plasmon excitation of AuNPs plays an essential role in the modulation of superconductivity. This is supported by our experimental observations, including (1) the T_c modulation coefficient follows the absorbance of AuNPs and reaches maximum close to the plasmon resonance peak (see Fig. R14k), (2) the rapid quenching of superconductivity modulation effect when increasing NbSe2 thickness to five layers (Fig. 1f in the main text),”.*

But the modulation of superconductivity demonstrated by the authors is in the time scale of a second and this cannot be seen as a rapid quenching of superconductivity.

Response: We thank the reviewer for the constructive comments.

The “rapid” here refers to the sensitive dependence on the thickness, not relevant to the time scale. To avoid confusion, we have deleted “rapid” in the revised manuscript.

Comment 10: *On page 24, the authors agree that their NbSe₂ film cannot be uniformly patterned in the substrate (evidenced by two T_c points in the transport), and later they mention that “the layer thickness indeed plays an essential role”, “Therefore, the electrodes are in direct contact with the bottom of the NbSe₂ flake”.*

This suggests that the authors have no control over the thickness of their devices, the number of layers might be more or less on different parts of the devices and so the device behaves differently depending on the illumination spot.

Response: We thank the reviewer for the valuable comments.

As explained in our last response, the two T_c in the transport is attributed to the nonuniform quality of the superconducting NbSe₂, not the thickness. Such nonuniformity is mainly due to the nanofabrication processes. In this work, the thickness of NbSe₂ layers is well controlled as confirmed by the optical contrast and atomic force microscopy (AFM).

Responses to Reviewer #3

General Comment: *This reviewer’s main question was whether the suppression of the superconductivity is due to plasmonic effect or the heating effect of the light. The authors answered that question by doing more measurements and updating the Fig.1b, Fig.S3 and adding new figures FigS3, FigS4, and FigS7. They also added the measurement for the wavelengths of 406 nm, 635 nm, and 730 nm. It is now more convincing to claim that the suppression of the superconductivity is solely due to plasmonic effect, instead of heating from the light. It is because at wavelength of 406 nm there is no suppression of superconductivity compared to plasmonic resonance wavelength of 532 nm, even though 406 nm carries more energy. This reviewer appreciates the authors’ hard work.*

This reviewer thinks this article will be interesting for the readers and should be published in Nature communications.

Response: We appreciate the reviewer’s recognition of the validity of our claim and its interest to the readers, as well as his recommendation for publication.

Comment 1: 1. By making the color scales of FigS7c and FigS7d same, it may be easier visually of readers to see the difference in on and off resonance.

Response: The data variation range of Fig. S7c and S7d are 0.708~0.712 and 0.0946~0.0948, respectively. So the contrast in both figures will be lost if unified with the same scale range. To avoid confusion, we have changed to different color scales, as shown in Fig. R2.

Figure R2. FDTD simulation results. (a) The device configuration for FDTD simulation. (b) The electric field strength distribution $(E/E_0)^2$ along the xz plane for the section of typical gold nanoparticle. (c,d) The electric field strength distribution $(E/E_0)^2$ along the xy plane close to the NbSe₂ layers (detection is set at $z=-6.5$ nm) for on resonance and off resonance cases.

Comment 2: Is it necessary to say “Few-Layer NbSe₂” in the title or “thin-film NbSe₂” will also work?

Response: We thank the reviewer for the constructive comments.

Following the reviewer’s suggestion, we have changed to “thin-film NbSe₂” and revised relevant phrases in the revised manuscript.

REVIEWERS' COMMENTS

Reviewer #1 (Remarks to the Author):

In this new round of comments, the authors did their best in addressing the remaining comments of Referee 2. To me, it is clear that the quenching of superconductivity requires both 1) AuNP to be present, 2) light excitation and the correct wavelength.

I agree with the authors that the evanescent field can couple to Cooper pairs via excitation of high-energy quasiparticles and phonons. Suppression of superconductivity does not require free electrons in the superconducting gap.

I further agree with the authors that finite elements simulations can be used to estimate the evanescent field in a metal. Introducing the superconductor in the model and expecting to observe the quenching of superconductivity would not be the correct approach in this case.

In conclusion, I still support publication of this paper in Nature Communications.

Reviewer #4 (Remarks to the Author):

I have read through the responses of the authors to the referees. One of the referees raises important questions. However, I believe that the authors have mostly addressed those comments. Moreover, Figures 1 (c) and (e) of the manuscript point toward the role of plasmon-mediated modulation.

I must highlight that the impact of plasmons is really small as can be seen in 1 (e) where the T_c shifts only by 1.2 K as the number of photons is increased. This is debatable but it is useful for superconducting switchable devices to have plasmon-assisted response for modulation to occur at lower photon fluences.

Overall, I support the author's data and the response presented. Such investigations are important for the field of superconductor optoelectronics in general and the experiments are challenging.